# Ubiquitin ligase ITCH regulates life cycle of SARS-CoV-2 virus

**Qiwang Xiang[1], Camille Wouters[2], Peixi Chang[3], Yu-Ning Lu[1], Mingming Liu[1], Haocheng Wang[1], Haley Heine[2], Sunning Qian[1], Junqin Yang[1], Andrew Pekosz[2], Yanjin Zhang[3], Jiou Wang[1,4]\***

[1]Department of Biochemistry and Molecular Biology, Bloomberg School of Public Health, Johns Hopkins University, Baltimore, United States; [2]Department of Molecular Microbiology and Immunology, Bloomberg School of Public Health, Johns Hopkins University, Baltimore, United States; [3]Department of Veterinary Medicine, University of Maryland, College Park, United States; [4]Department of Neuroscience, School of Medicine, Johns Hopkins University, Baltimore, United States

## eLife Assessment

This **valuable** study demonstrates that the E3 ligase ITCH regulates several steps of the SARS-CoV-2 replication cycle by enhancing ubiquitination of viral envelope and membrane proteins. The phenotypic data are based on **solid** evidence showing a role for ITCH in distinct phases of viral replication and host processes. The findings lay the ground work for future studies to decipher detailed molecular mechanisms that explain how ITCH regulates SARS-CoV-2.

**\*For correspondence:**
jiouw@jhmi.edu

**Competing interest:** The authors declare that no competing interests exist.

**Abstract** Severe acute respiratory syndrome coronavirus 2 (SARS-CoV-2) infection poses a major threat to public health, and understanding the mechanism of viral replication and virion release would help identify therapeutic targets and effective drugs for combating the virus. Herein, we identified E3 ubiquitin protein ligase Itchy homolog (ITCH) as a central regulator of SARS-CoV-2 at multiple steps and processes. ITCH enhances the ubiquitination of viral envelope and membrane proteins and mutual interactions of structural proteins, thereby aiding in virion assembly. ITCH-mediated ubiquitination also enhances the interaction of viral proteins to the autophagosome receptor p62, promoting their autophagosome-dependent secretion. Additionally, ITCH disrupts the trafficking of the protease furin and the maturation of cathepsin L, thereby suppressing their activities in cleaving and destabilizing the viral spike protein. Furthermore, ITCH exhibits robust activation during the SARS-CoV-2 replication stage, and SARS-CoV-2 replication is significantly decreased by genetic or pharmacological inhibition of ITCH. These findings provide new insights into the mechanisms of the SARS-CoV-2 life cycle and identify a potential target for developing treatments for the virus-related diseases.

## Introduction

Severe acute respiratory syndrome coronavirus 2 (SARS-CoV-2), which has affected the lives of almost everyone around the world, is an enveloped, positive-sense, single-stranded RNA coronavirus (*Wang et al., 2020*; *Wu et al., 2020*). Despite the remarkable vaccination programs that protect people at high risk for complications, new infection cases are still occurring. As of February 2024, over 774 million people have been confirmed to be infected, and over 7.03 million confirmed deaths have been reported (https://covid19.who.int). Emerging evidence indicates that the virus can have long-lasting effects on various organ systems, a condition often referred to as long COVID (*Huang et al.,*

*2021*). It is estimated that at least 65 million individuals worldwide have long COVID, with the number of cases steadily rising (*Davis et al., 2023*; *Ballering et al., 2022*). The development of novel, effective treatment strategies that benefit public health relies on detailed knowledge of molecular and cellular mechanisms of viral replication and virion release.

The SARS-CoV-2 virion consists of the positive-sense RNA genome and four structural proteins: nucleocapsid protein (N), envelope protein (E), membrane protein (M), and spike protein (S). The genome-bound N proteins comprise the virion core, which is surrounded by the virion membrane containing E, M, and S proteins (*Cai et al., 2020*). The E protein, the smallest among the structural proteins, is integrated into the virion membrane and participates in virion assembly and budding (*Schoeman and Fielding, 2019*). The M glycoprotein, embedded in the virion membrane and being the most abundant structural protein, interacts with all the other structural proteins and plays an essential role in viral proliferation and immune evasion (*Thomas, 2020*; *Sui et al., 2021*; *Fu et al., 2021*). The S protein, which forms a homotrimeric complex in the virion, plays a critical role in viral entry, assembly, and release (*Cai et al., 2020*; *Hoffmann et al., 2020a*). Post-translational modifications (PTMs) of proteins are known to play a crucial role in protein functions.

One form of PTMs observed for virion structural proteins is ubiquitination, which plays essential roles in the budding, egress, and infection of retroviruses (*Joshi et al., 2008*), parainfluenza viruses (*Zhang et al., 2014*; *Harrison et al., 2012*), Nipah viruses (*Wang et al., 2010*), influenza A virus (*Su et al., 2018*), and Zika virus (*Giraldo et al., 2020*). Growing evidence suggests that SARS-CoV-2 structural proteins are also modified by ubiquitination, which may have functional consequences. E3 ligases NEDD4 and WWP1 were reported to promote S protein ubiquitination and viral egress (*Novelli et al., 2021*). RNF5-mediated ubiquitination of M facilitates the interaction between M and E and regulates virus-like particle (VLP) formation and release (*Yuan et al., 2022*). Furthermore, ubiquitinated peptides of other structural proteins (N, M, and E) were also observed through proteomic analysis (*Yuan et al., 2022*; *Zhang et al., 2021*; *Li et al., 2023*). However, it remains unclear how ubiquitination or other forms of posttranslational processing of these structural proteins influence the life cycle of these viruses.

Here, we identified the HECT ubiquitin E3 ligase ITCH as a central regulator of the functions of SARS-CoV-2 structural proteins. The functions of ITCH in the regulation of SARS-CoV-2 life cycle are mediated not only by the ubiquitination of the structural proteins but also by other cellular changes that alter the post-translational cleavage of specific structural proteins. ITCH was found to promote the ubiquitination of E and M, enhancing their interactions with other structural proteins. The ubiquitination of the SARS-CoV-2 structural proteins also promotes their release by targeting the viral proteins to the autophagosome as a secretory mechanism. Meanwhile, ITCH stabilizes S protein by suppressing the activities of S cleavage proteases via changes in several cellular processes. The critical role of ITCH in the life cycle of SARS-CoV-2 is confirmed by the significant decrease in the virus proliferation as a result of ITCH inhibition. Furthermore, ITCH is activated in host cells infected by SARS-CoV-2, suggesting that the virus can hijack this central regulator for its proliferation. Together, this study reveals critical and multi-faceted roles of the E3 ligase ITCH in the life cycle of SARS-CoV-2.

## Results

### ITCH ubiquitinates and interacts with the SARS-CoV-2 structural proteins E and M

S, E, and M proteins are sufficient for the formation of the SARS-CoV-2 infectious VLP (*Figure 1—figure supplement 1A*; *Mi et al., 2021*). To identify host proteins that interact with SARS-CoV-2 VLP components, we performed proteomic analysis of the co-immunoprecipitates pulled down by the structural proteins expressed in HEK293 cells (S/E/M, E, or M) (*Figure 1—figure supplement 1B*). The analysis revealed a group of proteins functioning in ubiquitination that interact with the structural proteins (*Figure 1—figure supplement 1C*, raw data details were included in materials and methods). Among the identified E3 ligases, ITCH had the highest number of peptides and was further confirmed by immunoprecipitation to interact with the structural proteins (*Figure 1—figure supplement 1D*).

To test the E3 ligase activity of ITCH on these structural proteins, Flag-tagged S/E/M proteins were co-expressed in HEK293 cells together with wild-type (WT) ITCH, or an inactivated form of ITCH (ITCH-CS) with the $C_{830}S$ mutation (*Marchese et al., 2003*), followed by ubiquitin immunoblotting analysis of

the Flag-immunoprecipitated S/E/M proteins under denaturing conditions. Compared with the inactive mutant control, ITCH promoted the ubiquitination of these structural proteins (*Figure 1—figure supplement 1E*). Subsequent analysis revealed the presence of Lys63 (K63) linkages in the polyubiquitin chains (*Figure 1—figure supplement 1F*), which mainly mediate protein–protein interactions or conformational changes (*Komander and Rape, 2012*). Analysis of the individual structural proteins in the presence of all three structural proteins indicated that ITCH did not regulate the ubiquitination of S (*Figure 1—figure supplement 1G*) but promoted the ubiquitination of both E and M (*Figure 1—figure supplement 1H, J*). The ubiquitination of E and M by ITCH involved K63-linked polyubiquitin chains (*Figure 1—figure supplement 1I, K*), and the ubiquitination of either E or M could occur in the absence of the other two structural proteins, suggesting that formation of S/E/M-containing VLPs was not required for ITCH-mediated ubiquitination of E or M (*Figure 1A, B*, *Figure 1—figure supplement 2A, B*). The role of ITCH in the ubiquitination of E and M was further verified by deleting ITCH in Vero-E6-TMPRSS2 (vT2) cells, a monkey cell line expressing TMPRSS2, a transmembrane protein critical for SARS-CoV-2 infection (*Matsuyama et al., 2020*; *Figure 1C*, *Figure 1—figure supplement 2C, D*).

To examine the association of ITCH with E or M, we performed ITCH immunoprecipitation assays and observed the pull-down of E or M (*Figure 1D*, *Figure 1—figure supplement 2E*). Reciprocal immunoprecipitation of E or M confirmed their interactions with endogenous ITCH (*Figure 1E*, *Figure 1—figure supplement 2F*). To provide evidence of the colocalization of the structural proteins and ITCH in situ, we performed the proximity ligation assay (PLA) in HEK293. As expected, strong PLA signals in the cytoplasm were observed using paired Flag (E or M) and ITCH antibodies, while little background signal was detected in the control groups (*Figure 1F*, *Figure 1—figure supplement 2G*). Lastly, the interaction between ITCH and E or M in virally infected cells was confirmed via endogenous E and M immunoprecipitation assays, and clear ITCH signals were observed in the E- and M-based precipitates (*Figure 1G*, *Figure 1—figure supplement 2H*). Furthermore, ITCH's role in the ubiquitination of E and M was substantiated in vT2 cells infected with SARS-CoV-2, where immunoprecipitation of endogenous E and M revealed a significant reduction in ubiquitinated E and M in ITCH-ablated cells (*Figure 1H*, *Figure 1—figure supplement 2I*).

## ITCH promotes the assembly of SARS-CoV-2 structural protein complexes

To investigate whether ITCH regulates virion assembly, we examined the effects of ITCH on the association of E or M with other structural proteins. First, to assess the association of E and M, we carried out immunoprecipitation assay using E as the bait to pull down M and found that ITCH did not alter the interaction between unmodified E and M (*Figure 2A*). Next, to examine the effect of ITCH on the interaction between E and S proteins, we conducted an affinity precipitation assay using agarose beads to capture tagged E protein as bait and measured the amount of co-precipitated S. Ectopic expression of ITCH led to an increase in the level of S (both intact and cleaved S proteins) co-precipitated by E, indicating that ITCH enhances the interaction between E and S (*Figure 2B*). Since ITCH promotes the ubiquitination of E, we examined the level of ubiquitinated E in the S/E complexes. We carried out serial precipitation assays using immunoprecipitation to isolate S protein-containing complexes, followed by the use of agarose beads to extract E protein from the precipitates obtained in the previous step under denaturing conditions (*Figure 2C*). Indeed, ubiquitinated E was found to be present in the S/E complexes, and the level was increased by ITCH (*Figure 2C*), suggesting that ITCH may enhance the interaction between E and S by promoting the E ubiquitination (*Figure 1—figure supplement 2J*).

To investigate the effect of ITCH on the association of M and S, we performed an immunoprecipitation assay using M as a bait to pull down S. Immunoblotting of the precipitates showed that the binding of M with S was not affected by ITCH (*Figure 2D*). In comparison when the association between M and E was examined, we found that the level of E immunoprecipitated by M was increased by ITCH, indicating that ITCH had a positive effect on the association between M and E (*Figure 2E*). Given that ITCH is associated with M and enhances its ubiquitination, it is possible that ITCH may promote the interaction of M and E via M ubiquitination (*Figure 1—figure supplement 2J*). The following serial precipitation assays were carried out using immunoprecipitation to pull down E protein-containing complexes and subsequently using agarose beads to pull down M protein from the obtained precipitates under denaturing conditions (*Figure 2F*). The results showed that ITCH

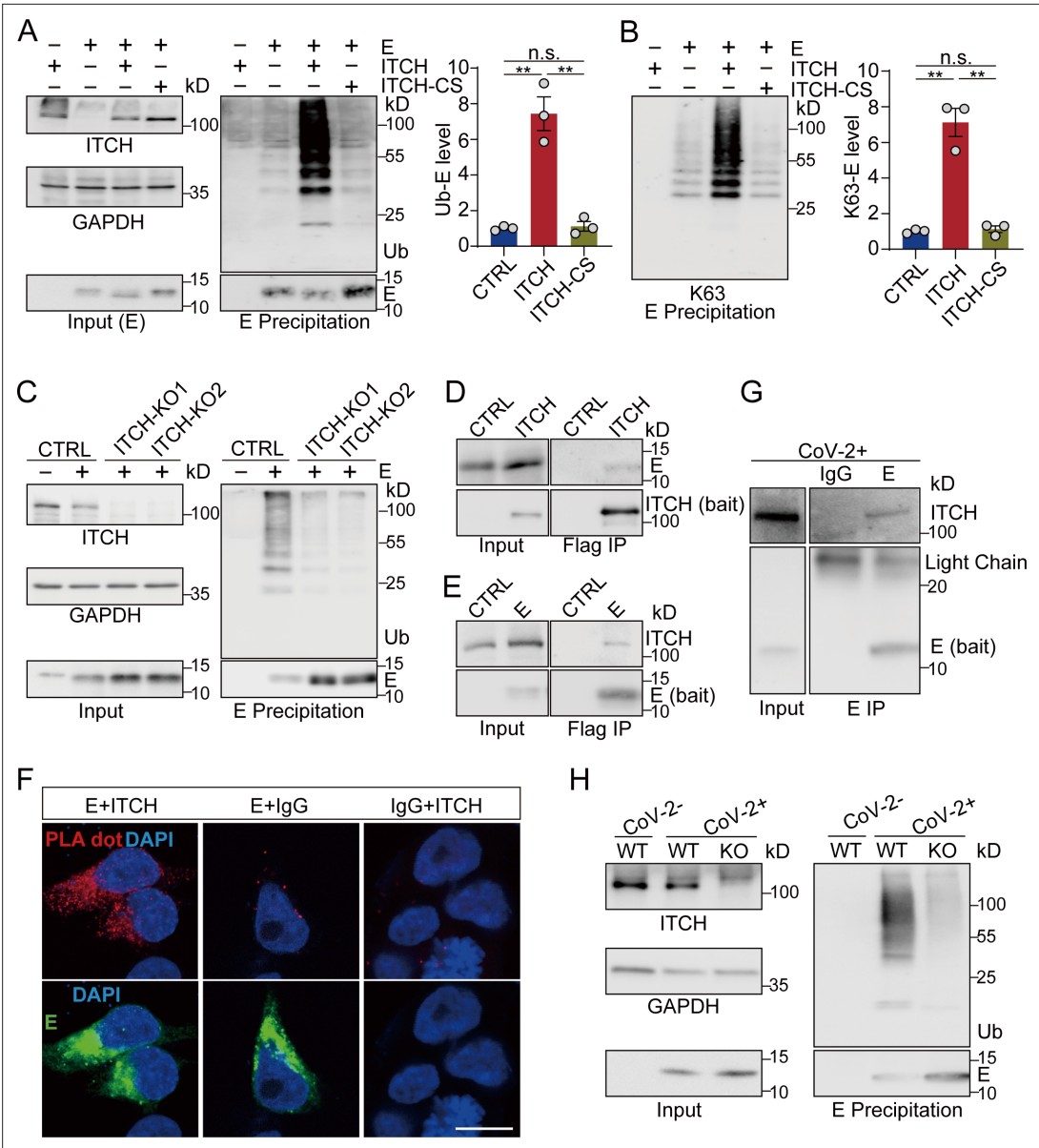

**Figure 1.** ITCH promotes the ubiquitination of the envelope protein E. (**A, B**) Denatured lysates from HEK293 cells expressing Flag-tagged E and HA-tagged ubiquitin with or without elevated ITCH or ITCH-CS were subjected to Flag IP. Immunoblotting of ubiquitinated E using ubiquitin antibody (**A**) or Ub-K63 antibody (**B**) showed that WT ITCH, but not the inactive ITCH-CS, promoted the ubiquitination of E through K63 polyubiquitin chains (*n* = 3). (**C**) Denatured lysates from vT2-WT and vT2-ITCH-KO cells expressing HA-tagged ubiquitin, Flag-2×Strep tagged E were subjected to Strep affinity precipitation (AP). Immunoblotting analysis showed a reduction of ubiquitinated E in the absence of ITCH. (**D**) Lysates from HEK293 cells expressing s-tag-fused E with Flag-tagged ITCH were subjected to Flag IP. (**E**) was present in ITCH's immunoprecipitates. (**E**) Lysates from HEK293 cells expressing Flag-tagged E were subjected to Flag IP, followed by immunodetection of coprecipitated endogenous ITCH. ITCH was detected in E precipitates. (**F**) Colocalization and interaction of ITCH with E was investigated in HEK293 cells by proximity ligation assay (PLA). Flag (rabbit) antibody was paired with ITCH (mouse) antibody. Negative controls comprised non-specific mouse IgG or rabbit IgG coupled with Flag or ITCH antibodies, respectively. Cells were visualized by confocal microscopy to detect red fluorescent dots, indicative of the colocalization of the respective antigens, targeted by the complementary species-specific antibodies. Scale bar, 10 μm. (**G**) Lysates from vT2 cells infected with SARS-CoV-2 at 1 MOI for 10 hr were subjected to E immunoprecipitation, followed by immunodetection using antibody against ITCH, revealing an ITCH signal in the E precipitates. (**H**) Denatured lysates from vT2-WT and vT2-ITCH-KO cells infected with SARS-CoV-2 at 1 MOI for 10 hr were subjected to E immunoprecipitation, and immunoblotting analysis revealed a reduction in ubiquitinated E in the absence of ITCH. Error bars represent means ± SEM. **p ≤ 0.01; n.s., non-significant.

The online version of this article includes the following source data and figure supplement(s) for figure 1:

**Source data 1.** PDF file containing original western blots for *Figure 1A–E, G, H*, indicating the relevant bands.

**Source data 2.** Original files for western blot analysis displayed in *Figure 1A–E, G, H*.

*Figure 1 continued on next page*

*Figure 1 continued*

**Figure supplement 1.** ITCH promotes ubiquitination of SARS-CoV-2 structural proteins.

**Figure supplement 1—source data 1.** PDF file containing original western blots for *Figure 1—figure supplement 1D–K*, indicating the relevant bands.

**Figure supplement 1—source data 2.** Original files for western blot analysis displayed in *Figure 1—figure supplement 1D–K*.

**Figure supplement 2.** ITCH interacts with M and promotes its ubiquitination.

**Figure supplement 2—source data 1.** PDF file containing original western blots for *Figure 1—figure supplement 2A–F, H, I*, indicating the relevant bands.

**Figure supplement 2—source data 2.** Original files for western blot analysis displayed in *Figure 1—figure supplement 2A–F, H, I*.

increased the level of ubiquitinated M in the E/M complexes (*Figure 2F*). Together, these data reveal that ITCH promotes the interactions among the SARS-CoV-2 structural proteins, likely through the augmentation of E/M ubiquitination mediated by ITCH.

## ITCH promotes the secretion of E and M proteins by facilitating their trafficking to autophagosome

Since ubiquitination plays a crucial role in viral protein secretion (*Isaacson and Ploegh, 2009*; *Valerdi et al., 2021*), we investigated the role of ITCH in regulating the secretion of E and M. We immunoprecipitated the secreted forms of these proteins from the culture media of both WT and ITCH knockout vT2 cells. We observed a significant decrease in the levels of E and M secreted from ITCH knockout cells compared to those from the WT cells (*Figure 3A*, *Figure 3—figure supplement 1A*), suggesting that ITCH promotes the secretion of these viral proteins. To understand how ITCH might influence the secretion of E and M, we analyzed the protein partners of these viral proteins identified in our proteomic analysis. Autophagy-related membrane protein 9A (ATG9A) was found to interact with SARS-CoV-2 structural proteins (*Figure 3—figure supplement 1B*). Many autophagy receptors, such as p62 (*Bjørkøy et al., 2005*), NBR1 (*Kirkin et al., 2009*), OPTN (*Richter et al., 2016*), NDP52 (*Ivanov and Roy, 2009*), and NIX (*Novak et al., 2010*), possess ubiquitin or ubiquitin-like modifier binding activity, which is required for the formation of autophagosomes, a process known to promote virion egress (*Yuan et al., 2022*). Therefore, we hypothesized that autophagy may be involved in ITCH-mediated secretion of E and M. To test this hypothesis, we performed precipitation assays of E and M followed by immunoblot analysis of various co-precipitated autophagic cargo receptors. Our results showed that the amount of the ubiquitin-associated autophagy receptor p62 co-precipitated with E increased in the presence of ectopically expressed ITCH and decreased in the absence of ITCH (*Figure 3B, C*). The levels of M-precipitated RTN3 and NDP52 were unchanged with or without the expression of ITCH or inactive ITCH-CS, whereas p62 precipitated by M was increased by ITCH (*Figure 3—figure supplement 1C*). Together, these data suggested that ITCH-ubiquitinated E and M interact specifically with the autophagic cargo receptor p62.

To examine whether the ITCH-dependent interaction between p62 and the viral proteins led to the formation of autophagosomes, we performed immunofluorescence analysis of p62 and the viral proteins for their cellular localizations. Both E and M exhibited marked colocalization with p62 in cellular puncta in the presence of ITCH but not the inactive ITCH-CS (*Figure 3D*, *Figure 3—figure supplement 1D*). To confirm that the increased p62-E or p62-M puncta corresponded to autophagosomes, we performed immunofluorescence analysis of the autophagosome-associated LC3B marker protein. The results showed increased fluorescent signals reflecting colocalization of E and M with LC3B in ITCH-overexpressing cells as compared to the control cells (*Figure 3E*, *Figure 3—figure supplement 1E*). Our data, taken together, indicate that ITCH enhances, in an E3 ligase-dependent manner, the autophagosome-associated formation of p62-E or p62-M complexes.

Further immunofluorescence assays were undertaken using an antibody against lysosome-associated LAMP1 to investigate the possible trafficking of ITCH-dependent p62-E/M complexes to lysosomes for degradation. The fluorescent signals of E and M showed little or no colocalization with LAMP1 even in the presence of overexpressed ITCH (*Figure 3—figure supplement 1F, G*), indicating that ITCH-dependent p62-E and p62-M complexes in autophagosomes are not targeted to the lysosome for degradation of the E and M.

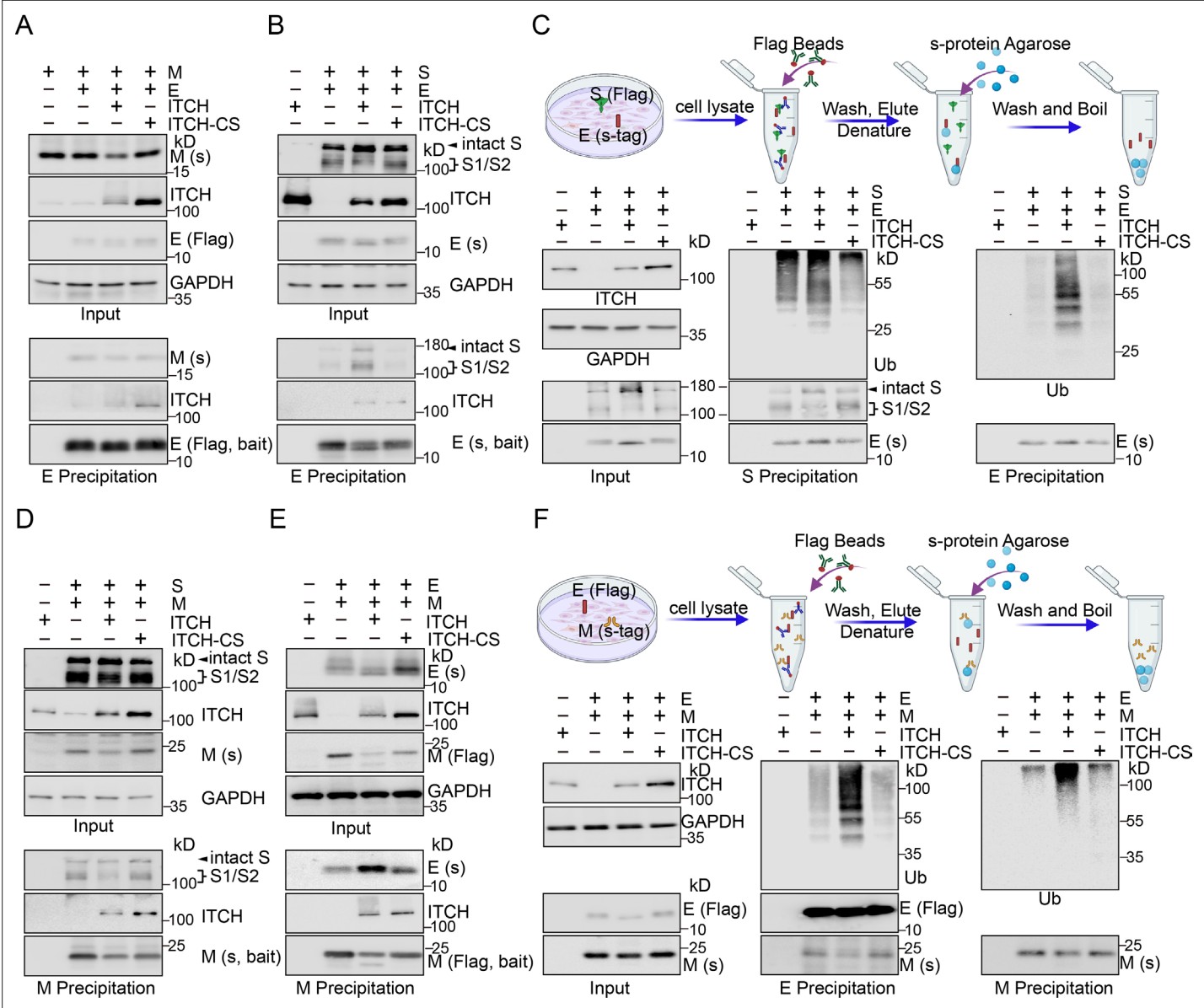

**Figure 2.** ITCH promotes mutual interactions of the structural proteins. (**A**) Lysates from HEK293 cells expressing Flag-tagged E and s-tag-fused M with ITCH or ITCH-CS were subjected to Flag IP, followed by immunoblotting for M. No significant change was observed in the interaction between E and unmodified M under these conditions. (**B**) S-tag AP was performed using lysates from HEK293 cells expressing s-tag-fused E and Flag-tagged S with ITCH or ITCH-CS. ITCH, but not the inactive ITCH-CS, increased the level of both intact and cleaved S precipitated by E. (**C**) Scheme of the serial precipitation assays isolating E protein from S containing complexes (top panel). Lysates from HEK293 cells expressing Flag-tagged S, s-tag-fused E, and ITCH or ITCH-CS together with HA-tagged ubiquitin were subjected to Flag IP, followed by s-tag AP under the denaturing condition. Immunoblotting analysis showed that S/E complexes contained ubiquitinated E and that the level of ubiquitinated E was significantly increased by ITCH. (**D**) Lysates from HEK293 cells expressing Flag-tagged S and s-tag-fused M with ITCH or ITCH-CS were subjected to s-tag AP, followed by immunoblotting for S. ITCH did not alter the interaction between M and S. (**E**) HEK293 cells expressing Flag-tagged M and s-tag-fused E with ITCH or ITCH-CS were subjected to Flag IP. Immunoblotting analysis showed that, while ITCH decreased the level of unmodified M, more E protein was precipitated by M. (**F**) Scheme of the serial precipitation assays isolating M protein from E containing complexes (top panel). Lysates from HEK293 cells expressing Flag-tagged E and s-tag-fused M with HA-tagged ubiquitin (Ub) and His6-fused ITCH or ITCH-CS were subjected to Flag IP, followed by s-tag AP under denaturing condition. Ubiquitinated M was observed in the E/M precipitates, and ITCH increased the level of ubiquitinated M in these complexes.

The online version of this article includes the following source data for figure 2:

**Source data 1.** PDF file containing original western blots for *Figure 2A–F*, indicating the relevant bands.

**Source data 2.** Original files for western blot analysis displayed in *Figure 2A–F*.

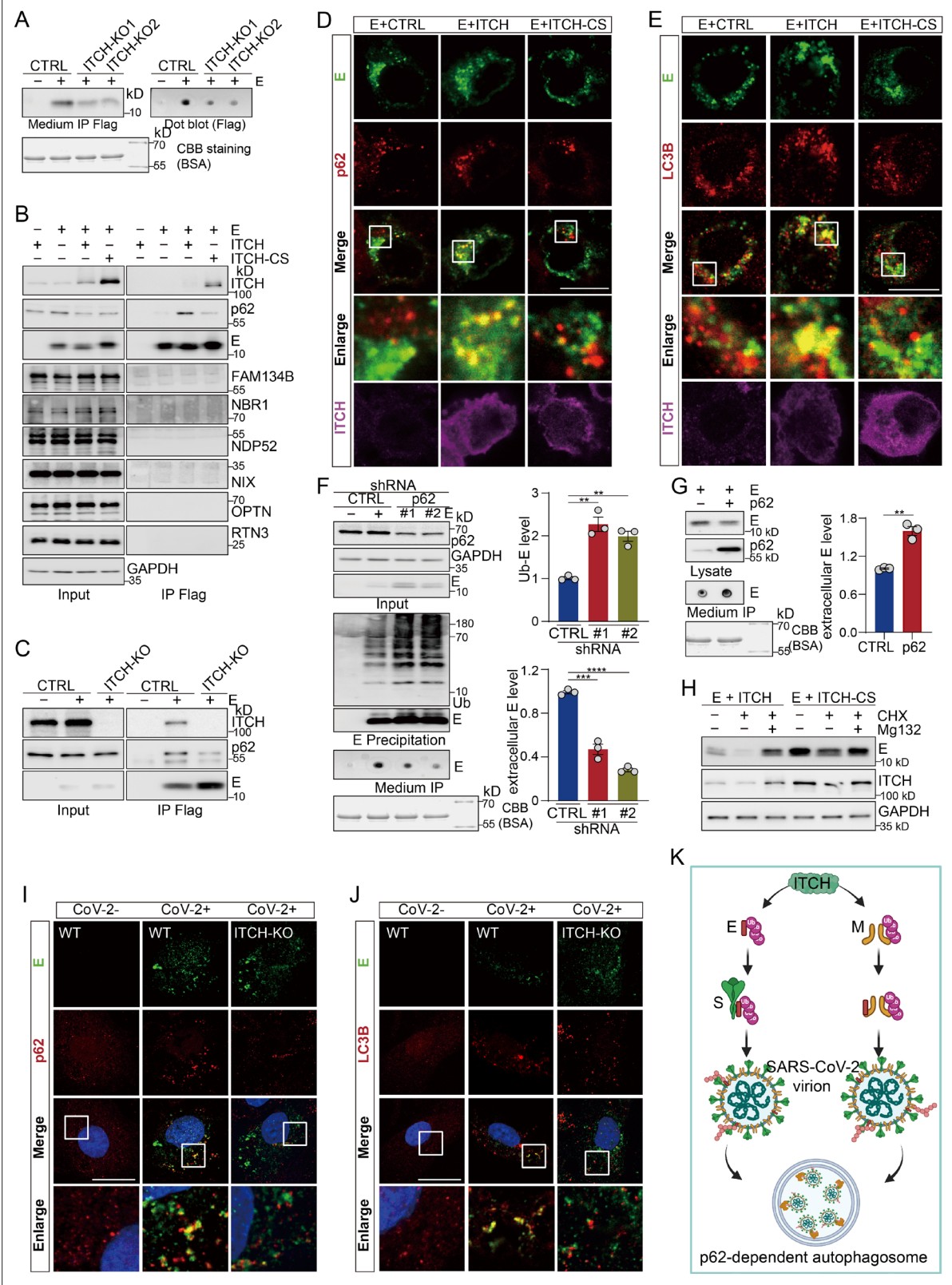

**Figure 3.** ITCH promotes the trafficking of E and M to p62-dependent autophagosomes, aiding secretion of E. (**A**) Culture medium from vT2-WT and vT2-ITCH-KO cells expressing HA-tagged ubiquitin (Ub) and Flag-2×Strep tagged E was harvested for Strep AP. A decrease of the extracellular E protein induced by ITCH ablation was visualized by immunoblot (left) and dot blot analysis (right). Coomassie Brilliant Blue (CBB) staining of the culture media is presented as a loading control. (**B**) Lysates from HEK293 cells expressing Flag-tagged E with ITCH or ITCH-CS were subjected to Flag

*Figure 3 continued on next page*

*Figure 3 continued*

IP, followed by immunoblotting for autophagosome cargo receptors. (**E**) specifically precipitated p62 and ITCH promoted their interaction. (**C**) Co-IP analysis of lysates from HEK293 WT and ITCH-KO cells expressing Flag-tagged E revealed that loss of ITCH reduces the specific interaction between M and p62. HEK293 cells were transfected with Flag-tagged E with ITCH or ITCH-CS. 24 hr later, cells were analyzed by immunofluorescence with Flag, ITCH and p62 or LC3B antibodies. ITCH enhanced the colocalization between E and p62 (**D**) or LC3B (**E**). Scale bar, 10 μm. (**F**) Culture media and denatured lysates from control (CTRL) and p62 knock down (#1, #2) HEK293 cells expressing HA-tagged ubiquitin (Ub), Flag-tagged E were subjected to Flag IP (with incorporation of a washing step with urea before elution for culture media samples), followed by immunoblotting or dot blot analysis. p62 depletion resulted in the accumulation of intracellular E (both unmodified and ubiquitinated), while decreasing the level of extracellular E (*n* = 3). (**G**) Culture media from HEK293 cells expressing Flag-tagged E together with control vector or p62 were harvested and subjected to Flag IP. The increase in extracellular E protein induced by ectopic p62 expression was detected by dot blot analysis (*n* = 3). CBB staining of the culture media is presented as a loading control. (**H**) HEK293 cells expressing E together with ITCH or ITCH-CS were cultured for 12 hr and then treated with CHX (10 μg/ml) and Mg132 (2 nM) or DMSO for 24 hr. Immunoblot analysis showed that MG132 blocked E degradation but did not rescue the ITCH-mediated reduction in E levels. (**I, J**) vT2-WT and vT2-ITCH-KO cells infected with SARS-CoV-2 at 1 MOI for 10 hr were subjected to immunofluorescence analysis with E and p62 or LC3B antibodies. ITCH-ablation decreased the colocalization between E and p62 (**G**) or LC3B (**H**). Scale bar, 10 μm. (**K**) A model of the function of ITCH in promoting autophagosome-mediated SARS-CoV-2 virion egress. ITCH-dependent ubiquitin modification enhances E binding with S and M binding with non-ubiquitinated E, resulting in the increase in virion formation and p62-dependent autophagosome targeting for release. Error bars represent means ± SEM. **p ≤ 0.01; n.s., non-significant.

The online version of this article includes the following source data and figure supplement(s) for figure 3:

**Source data 1.** PDF file containing original western blots for *Figure 3A–C, F–H*, indicating the relevant bands.

**Source data 2.** Original files for western blot analysis displayed in *Figure 3A–C, F–H*.

**Figure supplement 1.** ITCH promotes the trafficking of M from trans-Golgi network (TGN) to p62-dependent autophagosomes and enhances its secretion.

**Figure supplement 1—source data 1.** PDF file containing original western blots for *Figure 3—figure supplement 1A, C, H, I*, indicating the relevant bands.

**Figure supplement 1—source data 2.** Original files for western blot analysis displayed in *Figure 3—figure supplement 1A, C, H, I*.

To assess whether autophagosomes containing p62-E or p62-M complexes are involved in E/M secretion, we constructed cell lines with stable knockdown of p62 and then performed immunoprecipitation of E and M protein in lysates and culture media to measure their intracellular and secreted levels, respectively. Compared with those in the WT cells, both intracellular levels of ubiquitinated and non-modified E and M proteins were found to be increased in the p62 deficient cells, while extracellular E and M levels were significantly decreased (*Figure 3F*, *Figure 3—figure supplement 1H*). Moreover, E release was significantly enhanced upon ectopic expression of p62 (*Figure 3G*). To determine whether ITCH-dependent ubiquitination of E and M contributes to their degradation, we analyzed E and M protein levels in HEK293 cells coexpressing ITCH or the catalytically inactive mutant ITCH-CS and treated with cycloheximide (CHX; a protein synthesis inhibitor) and MG132 (a proteasome inhibitor). MG132 treatment blocked degradation of both E and M proteins, indicating that they can be degraded via the proteasome pathway. However, MG132 did not rescue the ITCH-mediated reduction in E and M levels, indicating that ITCH-dependent downregulation of E and M is not mediated through the proteasome pathway and further supporting a role for ITCH-mediated E/M ubiquitination in promoting their release (*Figure 3H*, *Figure 3—figure supplement 1I*). Finally, to confirm ITCH's role in mediating the colocalization of E with p62 or LC3B in virally infected cells, we performed immunofluorescence analysis of endogenous E and p62 or LC3B in WT and ITCH-KO cells infected with SARS-CoV-2. Consistently, ITCH ablation resulted in a significant decrease in the colocalization of E with p62 or LC3B (*Figure 3I, J*). A similar result was observed with the M protein colocalizing with p62 or LC3B in SARS-CoV-2 infected cells (*Figure 3—figure supplement 1J, K*). These data suggested that the E and M proteins, once ubiquitinated by ITCH, utilize autophagosomes for their secretion (*Figure 3K*).

## ITCH protects S protein from cleavage at proprotein convertase cleavage site by reducing trans-Golgi network-localized furin

Proteolysis of S protein (S cleavage) is critical in several stages of the viral life cycle, including viral entry, virion assembly, viral transmission, and cell-cell fusion (*Hoffmann et al., 2020a*; *Peacock et al., 2021*; *Jaimes et al., 2020*; *Zhao et al., 2021*; *Hoffmann et al., 2020b*; *Zang et al., 2020*; *Kishimoto et al., 2021*). We noticed that cells with heightened expression of ITCH exhibit much lower levels

of intracellularly cleaved S protein (*Figure 1—figure supplement 1E, G–I* and *Figure 2B–D*). Since ITCH is not involved in S protein ubiquitination, we asked whether ITCH regulates the intracellular cleavage process of the S protein, which produces the surface subunit S1 and the transmembrane subunit S2 (*Hoffmann et al., 2020a*). The cleavage of S protein occurs at a proprotein convertase cleavage (PPC) site, which harbors multiple arginine residues (*Figure 4A*). The analysis of S cleavage revealed that heightened expression of ITCH, but not the inactive ITCH-CS, led to an increase in the level of intact S protein, while the levels of the S1/S2 subunits were significantly reduced, indicating a negative effect of ITCH on S protein cleavage (ratio of cleaved S/intact S, *Figure 4B*). When the PPC site was mutated to render the S protein non-cleavable (*Figure 4A*), we further examined the effect of ITCH on S cleavage and secretion in vT2 cells. ITCH significantly increased the levels of full-length S in both intracellular and extracellular fractions while concomitantly reducing the levels of the S1 and S2 subunits. In contrast, the S MUT variant showed no changes in protein abundance or secretion in response to modulation of ITCH activity (*Figure 4D*).

To gain a better understanding of ITCH's role in S protein cleavage, we first tested whether ITCH interacts with the S protein. However, unlike with E and M proteins, we did not observe any ITCH signal in S immunoprecipitates (*Figure 4—figure supplement 1A*). Further immunofluorescence assays showed that there was no alteration in the subcellular localization of the S protein in cells ectopically expressing ITCH (*Figure 4—figure supplement 1B*). S can be cleaved by several host serine proteases or cysteine proteases, including furin (*Hoffmann et al., 2020a*; *Peacock et al., 2021*; *Jaimes et al., 2020*), cathepsin L (CTSL) (*Jaimes et al., 2020*; *Zhao et al., 2021*), transmembrane protease serine protease-2 (TMPRSS-2) (*Zang et al., 2020*; *Hoffmann et al., 2020b*), TMPRSS-4 (*Zang et al., 2020*), TMPRSS11D (*Kishimoto et al., 2021*), and trypsin (*Jaimes et al., 2020*; *Xia et al., 2020*). To determine whether ITCH affects the localization and expression of S-related proteases, vectors encoding C-terminally s-tag-labeled furin and N-terminally Flag-tagged versions of these proteases were constructed (*Figure 4—figure supplement 2A*). Immunofluorescence analysis revealed that the distribution patterns of furin (*Figure 4D*, *Figure 4—figure supplement 2B*) and CTSL (see below) were significantly altered by the expression of ITCH but not the inactive ITCH-CS, whereas subcellular localization of the other proteases did not show any significant changes (*Figure 4—figure supplement 2C–G*). Furin had a restricted and clustered distribution pattern in untreated cells; however, upon ITCH ectopic expression, the distribution of furin was transformed into a dispersed pattern in the cytoplasm, while the inactive ITCH-CS had no such effect (*Figure 4D*).

Intracellular furin is localized to the trans-Golgi network (TGN), where it cleaves the intracellular S protein into the S1 and S2 subunits (*Hoffmann et al., 2020a*; *Vey et al., 1994*; *Thomas, 2002*). To determine whether ITCH affects the TGN distribution of furin, we performed immunofluorescence assays. In control cells, furin was observed to colocalize with the TGN. However, heightened expression of ITCH but not the inactive ITCH-CS led to a significant decrease in TGN-localized furin signals (*Figure 4E*). To confirm the involvement of ITCH in regulating subcellular localization of furin in SARS-CoV-2 infected cells, we generated stable vT2 cells expressing flag-tagged furin. Following SARS-CoV-2 infection, immunofluorescence assays were conducted to visualize the subcellular localization of furin and TGN46. As expected, SARS-CoV-2 infection led to a marked reduction in TGN-localized furin signals; however, this reduction was significantly rescued in cells with ITCH ablation (*Figure 4F*). Further analysis showed that the protein level of furin was not affected by ITCH overexpression (*Figure 4—figure supplement 3A*) or ITCH ablation (*Figure 4—figure supplement 3H*). Additionally, the levels of TMPRSS and trypsin proteases were not affected by ITCH overexpression (*Figure 4—figure supplement 3B–F*) or ablation (*Figure 4—figure supplement 3I–M*). Together, these results indicate that ITCH does not change the expression or stability of these proteases but regulates the TGN localization of furin to influence S cleavage (*Figure 4G*).

## ITCH stabilizes S protein against degradation by inhibiting CTSL maturation

Among the S-associated proteases, CTSL efficiently degrades the S protein into smaller fragments after its initial cleavage by furin (*Jaimes et al., 2020*; *Zhao et al., 2021*). As mentioned above, ITCH also regulates the cellular distribution of CTSL. In untreated cells, ectopically expressing ITCH led to a disruption of the cytoplasmic distribution of CTSL, while the catalytically dead mutant ITCH-CS had no impact on its distribution (*Figure 5A*, *Figure 4—figure supplement 2H*). CTSL is synthesized

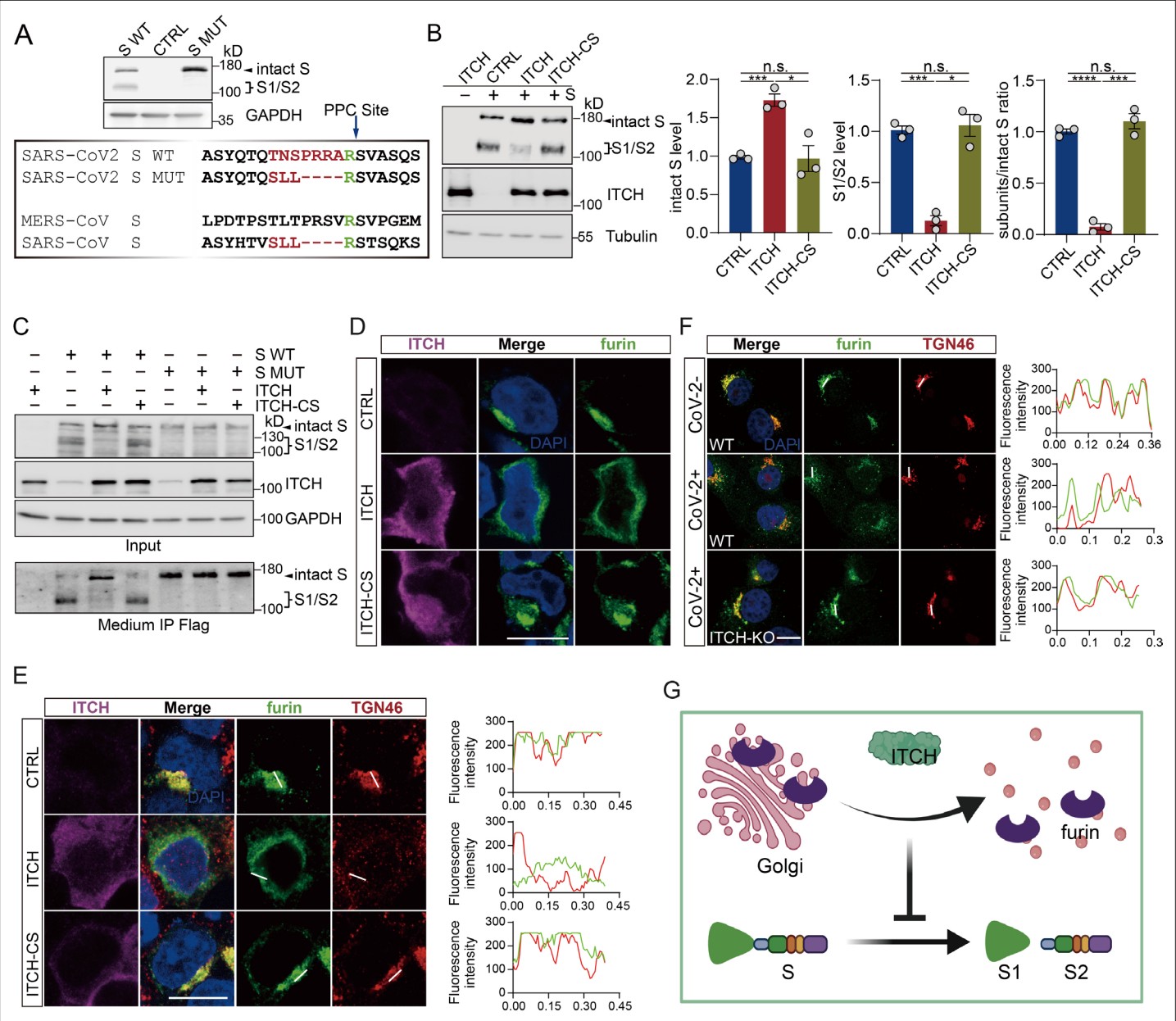

**Figure 4.** ITCH inhibits S cleavage by disrupting the Golgi localization of furin. (**A**) Sequence comparison of the S proteins from SARS-CoV-2, SARS-CoV, and MERS in a region containing the S1/S2 boundary. SARS-CoV-2 S contains the proprotein convertase cleavage (PPC) motif. The PPC site mutated from 'TNSPRRA' to 'SLL' (SARS-CoV) is highlighted in yellow. Immunoblotting analysis of wild-type S (S WT) and PPC site mutant (S MUT) proteins were shown using lysates from HEK293 cells expressing the variants. (**B**) Levels of intact S and proteolytically formed S1/2 subunits in lysates of HEK293 cells expressing Flag-tagged S with ITCH or ITCH-CS were analyzed by immunoblotting. ITCH significantly reduced the ratio of S1/2 subunits to intact S, indicating that ITCH inhibits S protein cleavage (*n* = 3). (**C**) Culture media from vT2 cells expressing Flag-tagged S WT (S WT) and S mutant (S MUT) proteins with ITCH or ITCH-CS were subjected to Flag IP (incorporating washing with urea before elution). Immunoblotting analysis showed that ITCH increased the levels of intact S in both intracellular and extracellular fractions, while decreasing the levels of S1/S2 subunits. The non-cleavable S MUT variant showed no changes in protein levels or secretion in response to modulations of ITCH activities. (**D**) Subcellular localization of s-tagged furin in HEK293 cells coexpressing ITCH or ITCH-CS was analyzed by furin and ITCH immunofluorescence staining. Elevated expression of ITCH, but not its inactive form, led to a dispersion of furin to the cytoplasm. Scale bar, 10 μm. (**E**) Transfected HEK293 cells expressing s-tagged furin with ITCH or ITCH-CS were analyzed via immunofluorescence staining using s-tag, TGN46, and ITCH antibodies. ITCH induced a dispersed of furin from trans-Golgi network (TGN) to the cytoplasm. Scale bar, 10 μm. (**F**) Subcellular localization of furin in vT2-WT and vT2-ITCH-KO cells with SARS-CoV-2 infection at 1 MOI for 10 hr was analyzed by furin and ITCH immunofluorescence staining. SARS-CoV-2 infection resulted in a pronounced dispersion of furin from the TGN to the cytoplasm, whereas ITCH ablation significantly inhibited this phenotype. Scale bar, 10 μm. (**G**) A model depicting ITCH's role in regulating

*Figure 4 continued on next page*

*Figure 4 continued*

the Golgi localization of furin and the related function of S cleavage. Error bars represent means ± SEM. *p ≤ 0.05; ***p ≤ 0.001; ****p ≤ 0.0001; n.s., non-significant.

The online version of this article includes the following source data and figure supplement(s) for figure 4:

**Source data 1.** PDF file containing original western blots for *Figure 4A–C*, indicating the relevant bands.

**Source data 2.** Original files for western blot analysis displayed in *Figure 4A–C*.

**Figure supplement 1.** ITCH does not associate with S and does not affect its subcellular localization.

**Figure supplement 1—source data 1.** PDF file containing original western blots for *Figure 4—figure supplement 1A*, indicating the relevant bands.

**Figure supplement 1—source data 2.** Original files for western blot analysis displayed in *Figure 4—figure supplement 1A*.

**Figure supplement 2.** ITCH does not affect subcellular localization of S-related TMPRSS and trypsin proteases.

**Figure supplement 2—source data 1.** PDF file containing original western blots for *Figure 4—figure supplement 2A*, indicating the relevant bands.

**Figure supplement 2—source data 2.** Original files for western blot analysis displayed in *Figure 4—figure supplement 2A*.

**Figure supplement 3.** ITCH does not affect the levels of TMPRSS and trypsin proteases.

**Figure supplement 3—source data 1.** PDF file containing original western blots for *Figure 4—figure supplement 3A–F, H–O*, indicating the relevant bands.

**Figure supplement 3—source data 2.** Original files for western blot analysis displayed in *Figure 4—figure supplement 3A–F, H–O*.

as a preproenzyme, which is processed into a proenzyme and subsequently a mature form (*Barrett and Kirschke, 1981*). We asked whether the ITCH-dependent distribution of CTSL was associated with the maturation of CTSL protein. In cells overexpressing ITCH, the proenzyme form (pro-form) of endogenous CTSL was increased relative to the levels seen in control cells, while the double-chain (DC) mature form was decreased (*Figure 5B*). The levels of CTSL were also assessed in WT and ITCH knockout cells. The pro-form of CTSL was nearly undetectable in the absence of ITCH, while the signal of the mature form was increased (*Figure 5C*). A similar analysis was performed with cells ectopically expressing CTSL for further confirmation. Consistent with the previous data, ITCH significantly increased the levels of the pro-form of CTSL and decreased the mature form (*Figure 4—figure supplement 3N*), indicating that ITCH inhibits CTSL maturation. To study the regulation of CTSL by ITCH under SARS-CoV-2 infection, we measured endogenous CTSL levels in vT2-WT and vT2-ITCH-KO cells with or without SARS-CoV-2 infection. We did not observe significant difference in CTSL protein levels between WT and ITCH-KO cells without SARS-CoV-2 infection. However, at 48 hours post-infection (hpi) with SARS-CoV-2, the level of the mature form of CTSL was significantly increased, while that of pro-CTSL was decreased in ITCH-KO relative to WT cells (*Figure 5D*), indicating that ITCH significantly inhibits CTSL maturation in SARS-CoV-2 infected cells.

Since ITCH inhibits CTSL maturation, we hypothesized that ITCH may provide protection against further cleavage of S1/S2. When ITCH and S1 or S2 subunits were co-expressed, a significant increase in the levels of S1 and S2 was observed, suggesting that ITCH enhances the stability of these two subunits (*Figure 5E*, *Figure 4—figure supplement 3O*). The level of non-cleaved S2 was increased by ITCH overexpression, while the formation of S2 fragments was significantly decreased (*Figure 5E*). Consistently, overexpression of ITCH led to lower levels of proteolytic cleavage products of the S protein (*Figure 5F*). For further confirmation, S cleavage in WT and ITCH-ablated cells with SARS-CoV-2 infection was detected. We found that both intact S and S subunits were decreased by ITCH ablation (*Figure 5G*). Taken together, our data show that in addition to its effect on furin, ITCH promotes CTSL trafficking and disrupts the processing of CTSL proenzyme, thereby protecting S and S1/S2 from further proteolysis (*Figure 5H*).

## Loss of ITCH suppresses SARS-CoV-2 virus proliferation

In view of the multiple important roles of ITCH in the processing of SARS-CoV-2 structural proteins, we postulated that ITCH could be a target for antiviral therapy. To test the effect of ITCH on the egress of native SARS-CoV-2 and the cytopathic effect (CPE) induced by SARS-CoV-2, we infected vT2-WT and vT2-ITCH-KO cells with SARS-CoV-2 at various multiplicities of infection (MOI). When vT2-WT cells were infected at an MOI of only 0.01, most cells died at 48 hpi, while progressively less cell death was observed at MOIs of 0.001 and 0.0001. ITCH ablation effectively inhibited CPE at all three MOIs (data not shown). Next, we measured the virus infectious titer and copy number in the culture medium

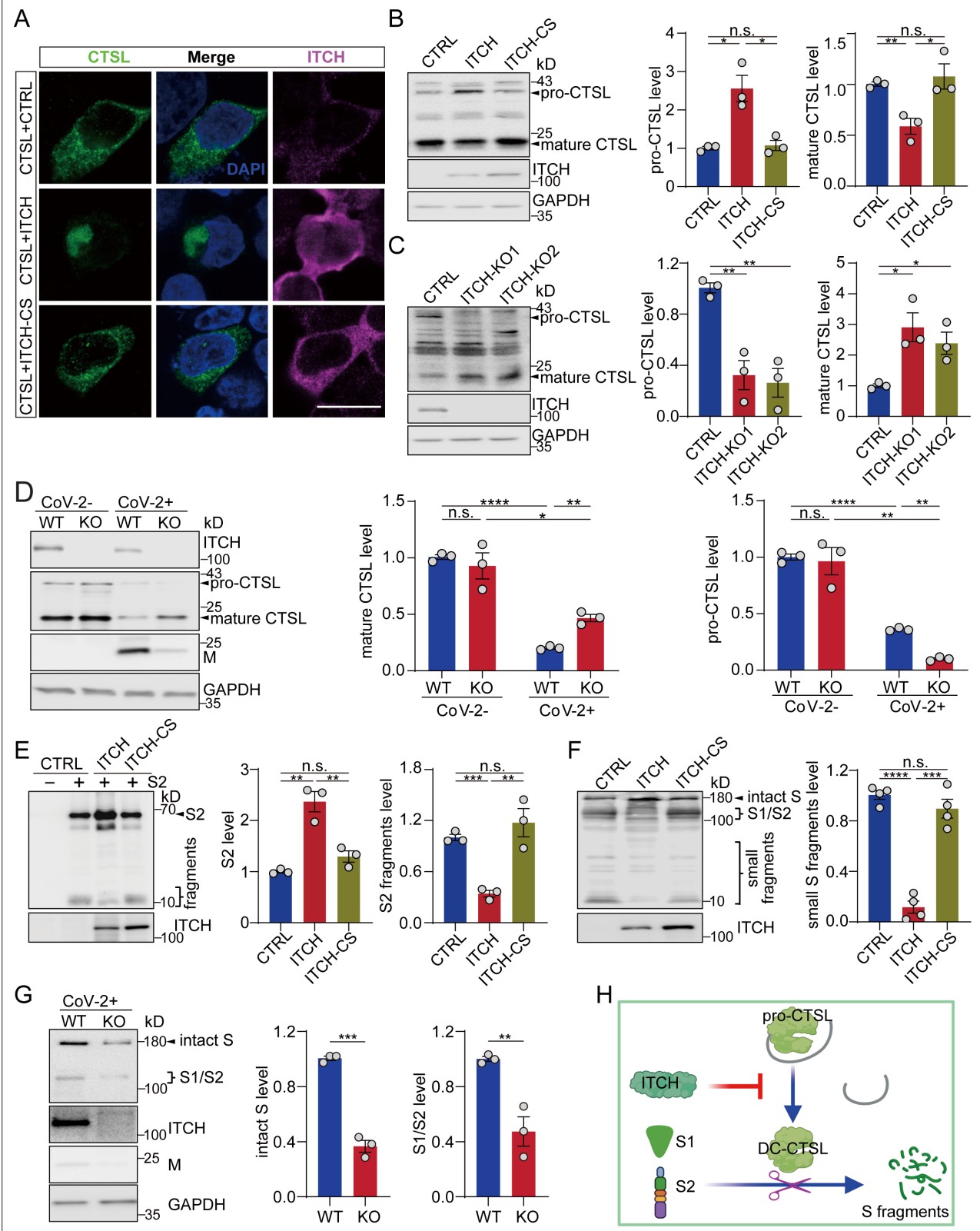

**Figure 5.** ITCH promotes the trafficking and maturation of CTSL and protects S protein from degradation. (**A**) Subcellular localization of Flag-tagged CTSL in HEK293 cells expressing ITCH or ITCH-CS was analyzed by Flag and ITCH staining. ITCH disrupted the cytoplasmic distribution of CTSL. Scale bar, 10 µm. (**B**) Immunoblotting analysis of endogenous CTSL in HEK293 cells expressing ITCH or ITCH-CS showed that ITCH increased the level of pro-CTSL while decreasing that of mature CTSL (*n* = 3). (**C**) Immunoblotting analysis of endogenous CTSL in WT and ITCH-KO HEK293 cells showed that

*Figure 5 continued on next page*

*Figure 5 continued*

ITCH ablation promoted CTSL maturation (*n* = 3). (**D**) Immunoblotting analysis of CTSL levels in vT2-WT and vT2-ITCH-KO cells infected with SARS-CoV-2 at a MOI of 0.0001 and harvested at 48 hpi. An increase in CTSL maturation was observed upon ITCH ablation, as indicated by the increased ratio of mature to pro-CTSL, under the condition of SARS-CoV-2 infection (*n* = 3). (**E**) Immunoblotting analysis of the S2 subunit and the derived S2 sub-fragments from HEK293 cells expressing S2 and ITCH or ITCH-CS showed that ITCH increased the level of uncleaved S2 while decreasing the levels of S2 sub-fragments (*n* = 3). (**F**) Immunoblotting analysis of small proteolytic fragments of S protein in HEK293 cells expressing Flag-tagged S and ITCH or ITCH-CS showed that ITCH reduced the levels of the S proteolytic cleavage fragments (*n* = 3). (**G**) Immunoblotting analysis of intact S and proteolytically formed S1/2 subunits from vT2-WT and vT2-ITCH-KO cells with SARS-CoV-2 infection (0.0001 MOI) at 48 hpi showed that ITCH ablation decreased the levels of both intact S and S1/2 subunits (*n* = 3). (**H**) A model depicting the function of ITCH in inhibiting CTSL maturation and subsequently reducing S cleavage. Error bars represent means ± SEM. *p ≤ 0.05; **p ≤ 0.01; ***p ≤ 0.001; ****p ≤ 0.0001; n.s., non-significant.

The online version of this article includes the following source data for figure 5:

**Source data 1.** PDF file containing original western blots for *Figure 5B–G*, indicating the relevant bands.

**Source data 2.** Original files for western blot analysis displayed in *Figure 5B–G*.

of vT2-WT and vT2-KO cells at 0.0001 MOI. Consistent with the observed CPE phenotype, SARS-CoV-2 production was strongly inhibited by ITCH ablation (*Figure 6A, B*). Notably, ITCH ablation reduced the viral copy number by approximately 8-fold (*Figure 6B*), while the infectious titer (TCID$_{50}$) decreased by at least 25-fold (*Figure 6A*), indicating that loss of ITCH markedly impairs the formation of infectious viral particles. This finding is consistent with the role of ITCH in altering the integrity of full-length S in SARS-CoV-2 virions (*Figure 6—figure supplement 1A*).

As phosphorylation of ITCH at T222 greatly enhances its enzymatic activity (*Gallagher et al., 2006*), we examined the level of phosphorylated ITCH (p-ITCH). Immunoblot analysis showed a robust increase in the level of p-ITCH in SARS-CoV-2-infected cells at 48 hpi despite a lower level of ITCH, possibly due to enhanced self-ubiquitination (*Figure 6C, D*), indicating that the SARS-CoV-2 infection significantly augmented the ITCH activity. Since the phosphorylation of ITCH-T222 is mediated by phosphorylated active JNK1 (p-JNK1) (*Gallagher et al., 2006*), we examined the level of p-JNK1 and found it significantly upregulated in both vT2-WT and vT2-ITCH-KO cells upon SARS-CoV-2 infection (*Figure 6C, E*). Therefore, the observed robust activation of ITCH in response to SARS-CoV-2 infection is mediated, in whole or part, by activation of JNK1. Next, we examined the level of p-ITCH during the course of SARS-CoV-2 infection. The level of phosphorylated ITCH did not show significant changes at the onset of infection (2 hpi) but exhibited an increase at 24 hpi, followed by a robust rise during the later stage of the viral life cycle (48 hpi) (*Figure 6F*). These findings indicate that the majority of ITCH exhibits weak activation during the initial virus infection stage but demonstrates robust activation during the replication stage of SARS-CoV-2.

To test the role of ITCH in SARS-CoV-2 proliferation and its potential as an antiviral target, we used a known ITCH inhibitor, clomipramine (Clom) (*Rossi et al., 2014*), for further analysis. First, we evaluated the minimal inhibitory concentration (MIC) of Clom in vT2 cells. vT2 cells with ectopic expression of ITCH and ubiquitin were treated with Clom at various concentrations, and the E3 ligase activity of ITCH was measured through the level of self-ubiquitination. Immunoblotting showed that, at the concentration of 15 µM, Clom significantly inhibited ITCH self-ubiquitination (*Figure 6—figure supplement 1B*). To test the effect of Clom on SARS-CoV-2 replication in vT2 cells, the drug was added to the culture medium 1 hr before SARS-CoV-2 infection. The CPE and virus titer were assessed and measured at 48 hpi. Clom-treated cells were substantially protected against SARS-CoV-2-induced CPE, and the virus copy numbers were reduced by thousands of folds in the presence of Clom (*Figure 6—figure supplement 1C*). As Clom inhibits multiple HECT E3 ligases, including ITCH (*Rossi et al., 2014*), the observed inhibitory effect on SARS-CoV-2 production may result from its broader inhibition of E3 ligases. We then tested Clom's effect on SARS-CoV-2 replication in vT2-ITCH-KO cells. In these cells, Clom strongly inhibited SARS-CoV-2-induced CPE as well (*Figure 6—figure supplement 1D*). However, unlike in wild-type vT2 cells, viral copy numbers were reduced by hundreds of folds (*Figure 6—figure supplement 1E*), suggesting that Clom's inhibition of SARS-CoV-2 production is partly due to its specific inhibition of ITCH E3 ligase activity.

Given that the lungs are the primary target of SARS-CoV-2 and lung failure is a leading cause of mortality in COVID-19 patients, we analyzed ITCH expression across various lung cell types. Using single-nucleus RNA sequencing data from a published database (*Melms et al., 2021*), which includes over 116,000 nuclei from the lungs of 19 individuals who died from COVID-19 and seven control

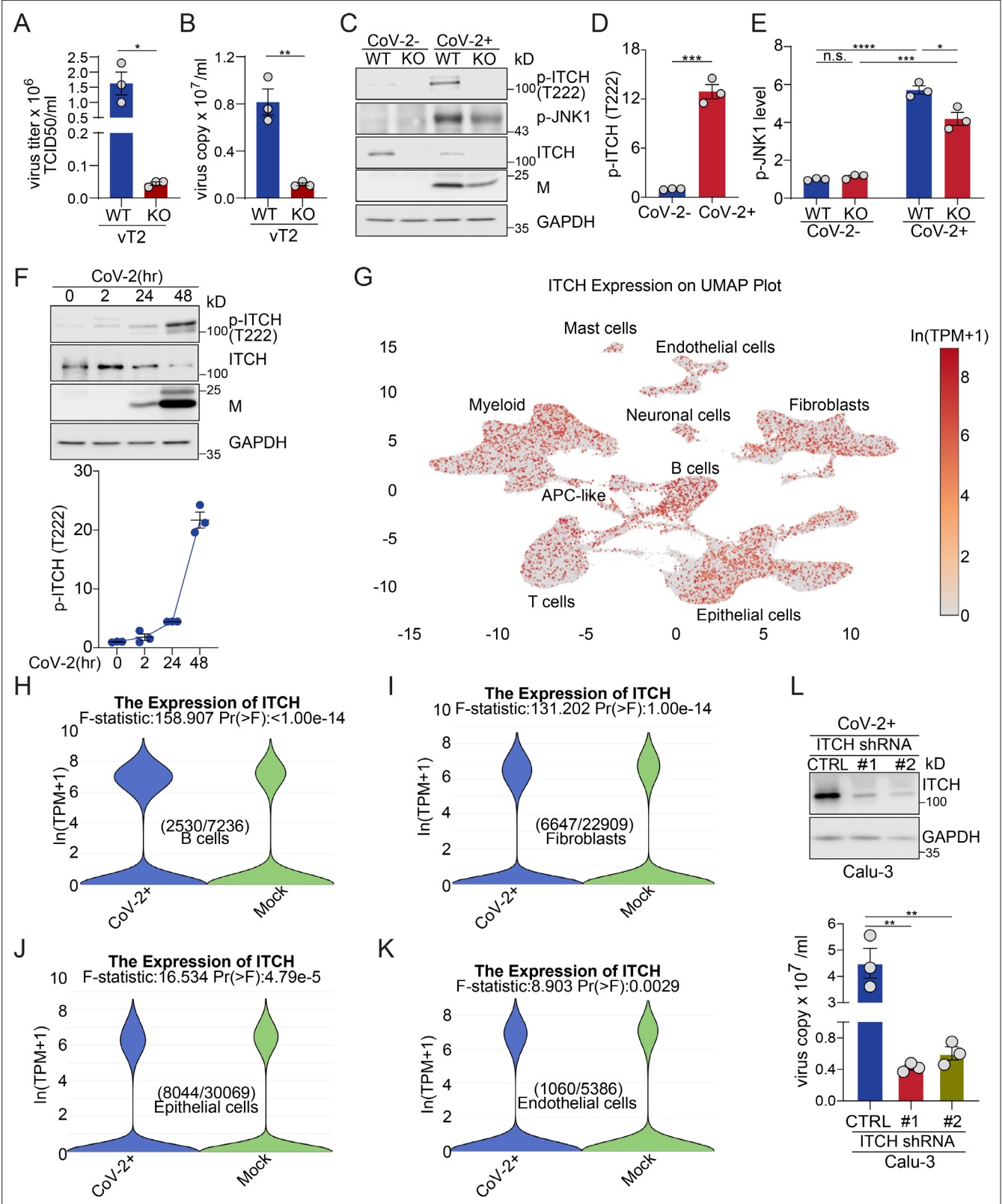

**Figure 6.** ITCH ablation suppresses SARS-CoV-2 production. (**A, B**) The analysis of infectious virus titers (TCID$_{50}$) and virus copy numbers in culture media of vT2-WT and vT2-ITCH-KO cells infected with SARS-CoV-2 at 0.0001 MOI, at 48 hpi, showed a robust inhibition of virus production by ITCH ablation (*n* = 3). (**C–E**) Immunoblotting analysis of phosphorylated ITCH (p-ITCH), ITCH and p-JNK1 in vT2-WT and vT2-ITCH-KO cells with SARS-CoV-2 infection at 48 hpi showed that the viral infection induced the activation of ITCH, as indicated by the significance increase in its phosphorylation (**C,**

*Figure 6 continued on next page*

*Figure 6 continued*

D) and the activation of JNK1, as indicated by the increase of p-JNK1 in both vT2-WT and vT2-ITCH-KO cells (C, E, *n* = 3). (**F**) Immunoblotting analysis of phosphorylated ITCH in vT2 cells, infected with SARS-CoV-2 at 0.01 MOI at various time points, showed time-dependent activation of ITCH, as indicated by its phosphorylation (*n* = 3). (**G–K**) Analysis of published single-nucleus RNA sequencing data from the lungs of 19 individuals who died from COVID-19 and seven control individuals showed that ITCH is ubiquitously expressed in major lung cell clusters (**G**) and that SARS-CoV-2 infection significantly elevates ITCH mRNA levels in B cells, fibroblasts, epithelial cells, and endothelial cells compared to controls. (**L**) The analysis of virus copy numbers in culture media of control and ITCH-depletion Calu-3-ACE2 cells infected with SARS-CoV-2 at 1 MOI for 48 hr revealed a significant reduction in virus production with ITCH depletion (*n* = 3). Error bars represent means ± SEM. *p ≤ 0.05; **p ≤ 0.01; ***p ≤ 0.001; ****p ≤ 0.0001; n.s., non-significant.

The online version of this article includes the following source data and figure supplement(s) for figure 6:

**Source data 1.** PDF file containing original western blots for *Figure 6C, F, L*, indicating the relevant bands.

**Source data 2.** Original files for western blot analysis displayed in *Figure 6C, F, L*.

**Figure supplement 1.** Pharmacological inhibition of ITCH suppresses SARS-CoV-2 production.

**Figure supplement 1—source data 1.** PDF file containing original western blots for *Figure 6—figure supplement 1A*, indicating the relevant bands.

**Figure supplement 1—source data 2.** Original files for western blot analysis displayed in *Figure 6—figure supplement 1A*.

individuals, we found that ITCH is ubiquitously expressed in lung cells (*Figure 6G*). Furthermore, SARS-CoV-2 infection significantly increases ITCH mRNA levels in specific lung cell types, including B cells, fibroblasts, epithelial cells, and endothelial cells (*Figure 6H–K*, *Figure 6—figure supplement 1F–J*). Recognizing that the ubiquitous expression and upregulation of ITCH in natural SARS-CoV-2 infection cells, we lastly examined viral replication under ITCH-depleted conditions in Calu-3-ACE2 cells, a human epithelial cell line engineered for SARS-CoV-2 research. Consistent with the phenotype observed in vT2-ITCH-KO cells, ITCH depletion in Calu-3 cells led to a significant decrease in viral copy number in the culture medium compared to wild-type cells (*Figure 6L*). Altogether, our data show that ITCH positively regulates SARS-CoV-2 replication and represents a potentially useful target for antiviral therapy and suppression of the infectivity of SARS-CoV-2.

## Discussion

In the present study, we have identified the E3 ubiquitin ligase ITCH as a central regulator influencing various stages and processes of SARS-CoV-2 (*Figure 7*). The functions of ITCH in the regulation of SARS-CoV-2 virion assembly, egress, and virus proliferation are mediated not only by the ubiquitination of the structural proteins but also by the post-translational cleavage of specific structure proteins. We found that ITCH promotes the ubiquitination of E and M, enhancing their interactions with other structure proteins, which would aid in virion assembly. The ubiquitination of the SARS-CoV-2 structural proteins also promotes the release of virion by targeting the viral proteins to the autophagosome as a secretory mechanism. Furthermore, ITCH stabilizes the S protein by inhibiting the activities of S cleavage proteases, including furin and CTSL. Notably, ITCH is activated in host cells by SARS-CoV-2 infection, suggesting that the virus can hijack this central regulator for its proliferation. Indeed, genetic or pharmacological inhibition of ITCH significantly decreased the replication of SARS-CoV-2, indicating that ITCH may be a promising target for the development of treatments for the viral disease.

PTMs such as ubiquitination may play an important role in the life cycle of coronaviruses. Here we identified that the E3 ligase ITCH associates with SARS-CoV-2 structural proteins and promotes their ubiquitination and mutual interactions, thereby enhancing the virion assembly. In addition, ITCH-dependent ubiquitination of E and M is necessary for their recognition by p62 and subsequent sequestration into autophagosomes. Interestingly, the enhanced sequestration of E and M proteins to the autophagosomes did not lead to increased localization of these proteins to the lysosome but facilitated the secretion of these structural proteins. Autophagosomes have been observed to mediate exosome secretion as they fuse with multi-vesicular bodies for intraluminal vesicle secretion (*Cheng et al., 2018*; *Xu et al., 2018*; *Leidal et al., 2020*). Accordingly, p62, the autophagosome ubiquitin receptor, plays a crucial role in promoting the ITCH-dependent secretion of E and M proteins, which ultimately facilitates virion egress. These findings reveal a previously undescribed mechanism of SARS-CoV-2 virion assembly and egress and provide new insights into the function of ubiquitination of viral proteins.

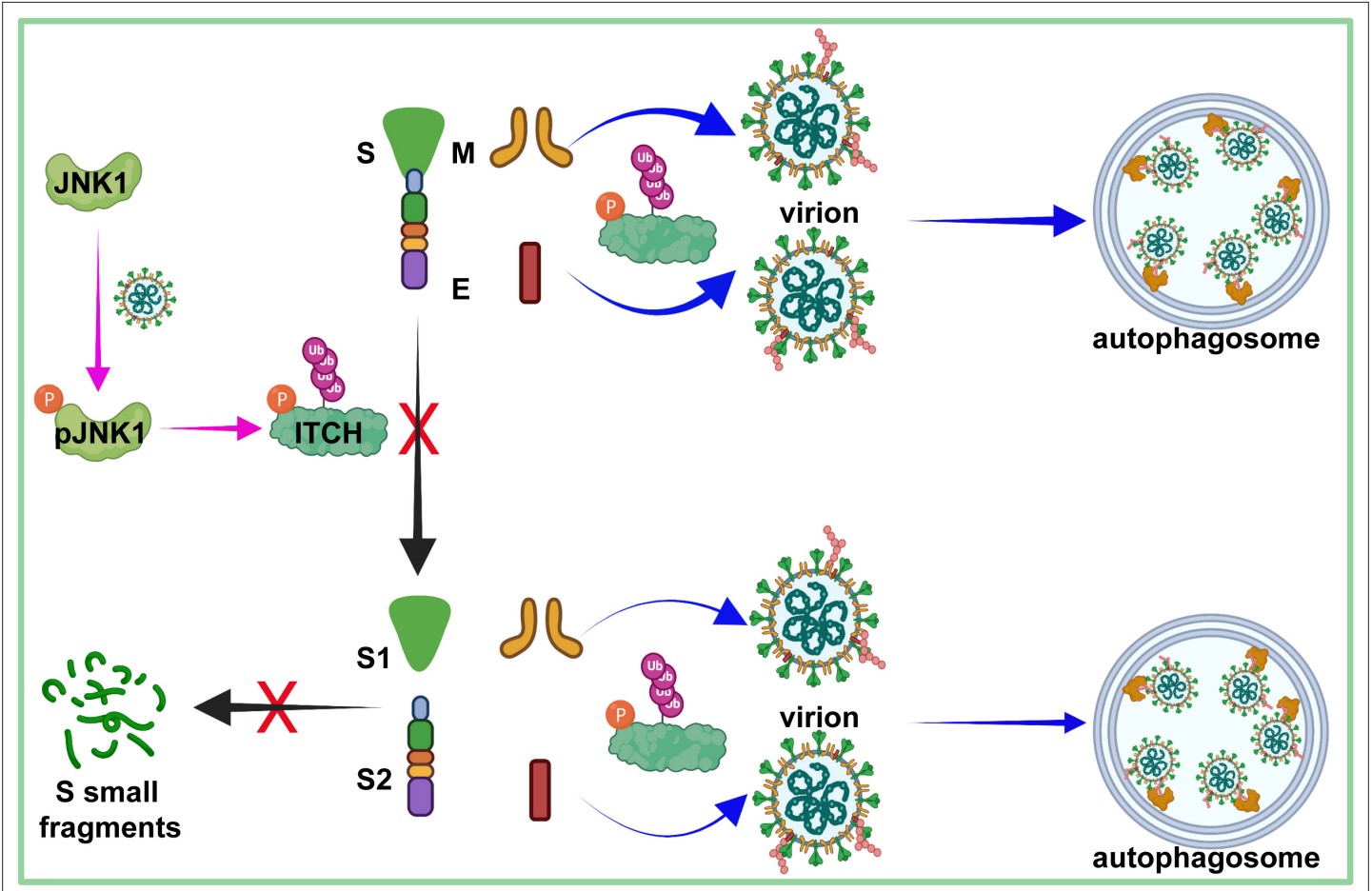

**Figure 7.** A model for the multiple roles of ITCH in the SARS-CoV-2 life cycle. The activation of JNK1 occurs during the late stage of the SARS-CoV-2 life cycle, which in turn activates the E3 ligase activity of ITCH (pink arrows). ITCH promotes ubiquitination of the E and M proteins, resulting in increased virion formation and p62-dependent autophagosome targeting (blue arrows), aiding SARS-CoV-2 egress. ITCH disrupts furin and CTSL protease activities, resulting in increased incorporation of intact S protein into the virion and enhanced virion infectivity and stability (black arrows).

In addition, our studies showed that ITCH does not regulate S ubiquitination but instead inhibits S proteolytic cleavage by reducing TGN-associated furin and inhibiting CTSL maturation. Although the S1/S2 cleavage is necessary for SARS-CoV-2 infection, recent studies suggest that enhanced cleavage may be disadvantageous for the virus. For instance, in Vero cells lacking TMPRSS2, virions with cleaved S1 and S2 are less stable (*Peacock et al., 2021*). Moreover, the D614G substitution, which renders the S protein more resistant to cleavage, reduces the release of the S1-domain and increases the incorporation of intact S protein into the virion, leading to enhanced virion infectivity and stability (*Zhang et al., 2020*; *Plante et al., 2021*; *Daniloski et al., 2021*). Therefore, a high degree of intracellular S cleavage plays a negative role in the SARS-CoV-2 life cycle. In the present study, we observed that ITCH inhibits S cleavage by reducing TGN-associated furin. This finding suggests that ITCH may promote the production of infectious virions during the SARS-CoV-2 life cycle via inhibition of S cleavage.

We also observed that ITCH inhibits the maturation of CTSL, which cleaves S into smaller fragments. Although the CTSL-dependent S cleavage facilitates the release of the SARS-CoV-2 genome from the endosome into the cytosol following endocytosis (*Jaimes et al., 2020*; *Zhao et al., 2021*), being a lysosomal protease, it serves as a mechanism to limit SARS-CoV-2 egress as β-coronaviruses exploit lysosomal trafficking for egress (*Ghosh et al., 2020*). Overall, during the virus replication stage, the inhibition of CTSL maturation by ITCH would promote the assembly and egress of infectious SARS-CoV-2 virions. Additionally, ITCH is found to protect targets of certain proteases from proteolytic degradation. Multiple virus families, including Borna-, Bunya-, Corona-, Filo-, Flavi-, Herpes-,

Orthomyxo-, Paramyxo-, Pneumo-, Retro-, and Togaviridae, have been reported to utilize furin for envelope glycoprotein cleavage, a crucial step in their viral life cycle (*Braun and Sauter, 2019*). In addition, CTSL-mediated proteolysis has been observed for envelope glycoproteins, such as the S protein of SARS/MERS-CoV and the GP protein of Ebolavirus (EBOV) (*Zhou et al., 2016*; *Simmons et al., 2005*). Taken together, ITCH may play a significant role in the viral life cycles of these viruses.

After the SARS-CoV-2 infection stage, the expression of p-JNK1 is induced. As one important activator of ITCH (*Gallagher et al., 2006*), the induction of p-JNK1 would enhance the E3 ligase activity of ITCH. Indeed, the activation of ITCH is low in the absence of SARS-CoV-2 infection. Consistently, without SARS-CoV-2 infection, ITCH ablation does not exhibit involvement in the maturation of CTSL processing in vT2 cells, further implying that ITCH does not affect the CTSL-dependent virus infection process. After the stage of SARS-CoV-2 infection, ITCH-dependent inhibition of CTSL maturation occurs, indicating a significant increase in the E3 ligase activity of ITCH that corresponds with the induction of p-JNK1 and the activation of p-ITCH at the T222 site. Further research is needed to fully understand the mechanisms underlying the induction in p-JNK1 and its potential impact on the immune response and overall clinical outcomes in COVID-19 patients.

The knockout of ITCH in vT2 cells inhibited SARS-CoV-2 production by 10- to 20-fold. Interestingly, the presence of the ITCH inhibitor Clom reduced virus production by a thousand-fold. It is possible that the observed inhibitory effect is partially due to drug cytotoxicity. Clom has the ability to inhibit other HECT E3 ligases besides ITCH (*Rossi et al., 2014*), and there is evidence that other HECT-type E3 ligases, such as NEDD4 and WWP1, play important roles in promoting S ubiquitination and virus egress (*Novelli et al., 2021*). Therefore, it is possible that the observed inhibitory effect of Clom on SARS-CoV-2 production may be due to its inhibition of multiple HECT-type E3 ligases, including ITCH, NEDD4, and WWP1.

Based on the data obtained in this study, we propose a model for the multiple activities of ITCH related to the SARS-CoV-2 life cycle (*Figure 7*). In the separate studies (*Xiang et al., 2025*; *Xiang et al., 2026*), we found that ITCH is abnormally accumulated in the tissues of patients suffering from neurodegenerative diseases such as Alzheimer's disease and amyotrophic lateral sclerosis. Importantly, ITCH acts as a critical regulator of Golgi complex integrity, lysosomal protein trafficking, and proteotoxicity associated with these disorders. These observations provide a mechanistic basis for ITCH-mediated alterations in furin subcellular localization and CTSL maturation (Golgi fragmentation). Increasing evidence links SARS-CoV-2 infection to neurodegenerative diseases (*Abu-Abaa et al., 2023*; *Li and Bedlack, 2021*; *Li et al., 2022*; *Frontera et al., 2022*; *Cilia et al., 2020*; *Lee et al., 2021*; *Matias-Guiu et al., 2021*). Given the dual role of ITCH in SARS-CoV-2 replication and neurodegeneration, the findings of the present study offer a foundation for developing antiviral and therapeutic strategies to mitigate SARS-CoV-2 infection and its associated pathological consequences.

# Materials and methods
## Plasmids

Lentiviral vectors pCEBZ-S-SF, pCEBZ-E-SF, pCEBZ-M-SF, pCEBZ-E-S, pCEBZ-M-S, pCEBZ-E-CBD, and pCEBZ-M-CBD expressing C-terminus tagged proteins were generated by PCR amplification of the respective sequences, from plasmid pCMV14-3X-Flag-S (Addgene #145780, cd21624), pLVX-E-2×Strep-IRES-Puro (Addgene #141385, Wuhan-Hu-1), pLVX-M-2×Strep-IRES-Puro(Addgene #141386, Wuhan-Hu-1) with NotI/BamHI-, or BsmBI-flanked primers and replacement of RFP coding sequences in pCEBZ-RFP-SF, pCEBZ-RFP-S, pCEBZ-RFP-CBD which are gifts from John Nicholas (*Chen et al., 2017*; *Chen and Nicholas, 2015*). The vector pCDNA3.n3xFlag-DEST-S expressing N-terminus and C-terminus Flag-tagged S was generated using the Gateway cloning system (Thermo Fisher) with primers containing a Flag tag coding sequence. A lentiviral vector of an S protein variant (PPC motif mutant) was generated by overlap PCR and cloned into the pCEBZ-RFP-SF. Lentiviral vectors expressing S subunits were generated by PCR and cloned into the pCEBZ-RFP-SF. The lentiviral vectors pLU-CMV-His-ITCH and pLU-CMV-Flag-ITCH were generated by replacing TRAF6 coding sequences in pLU-CMV-TRAF6 (Addgene #66923) with a NheI/SalI-flanked sequence which was overlap PCR amplified from a cDNA library (HeLa, Clontech, catalog number 638862) using appropriate custom primers. The lentiviral vector pLU-CMV-His-ITCH-CS was generated by insertion of a modified ITCH sequence (coding sequence with a Cys>Ser point mutation generated by overlap

PCR) into the corresponding cloning site of pLU-CMV-TRAF6, thereby replacing the TRAF6 sequence. The vector pRK5-HA-Ubiquitin encoding HA-tagged ubiquitin was obtained from Addgene (#17608). The lentiviral vectors pCEBZ-Flag-furin, pCEBZ-Flag-CTSL, pCEBZ-Flag-TMPRSS2, pCEBZ-Flag-TMPRSS4, pCEBZ-Flag-TMPRSS11D, pCEBZ-Flag-GST-ACR, and pCEBZ-Flag-GST-PLAT expressing various N-terminally Flag-tagged proteases were generated by PCR amplification of the respective sequences from the plasmids pDONR223-furin (Addgene #82122), p-hCathepsin L (Addgene #11250), pCSDest-HA-TMPRSS2 (Addgene #154963), p-MAC-TMPRSS4 (Addgene #172422), pDONR221-TMPRSS11D (Addgene #158447), pHH0103-ACR-Trypsin (Addgene #110042), and pHH0103-PLAT-Trypsin (Addgene #110039) using NotI/SalI-flanked primers (with stop codons), thereby replacing the RFP coding sequence in pCEBZ-RFP-SF. The pCEBZ-furin-S vector expressing furin with a C-terminal s-tag was generated by replacing the RFP coding sequence in pCEBZ-RFP-S with a NotI/SalI-flanked sequence PCR amplified from pDONR223-furin with primers without stop codon.

## Cell culture and transfection

Vero (ATCC, CCL-81) and vT2 cells were grown in Dulbecco's modified Eagle's medium (DMEM) supplemented with 10% fetal bovine serum (FBS), 1 mM sodium pyruvate and 1X GlutaMAX Supplement. HEK293 cells were cultured in DMEM supplemented with 10% FBS and 10 µg/ml gentamicin. HEK293 cells (ATCC, CRL-3216) at approximately 40–60% confluence were transfected with mixtures of plasmid DNA and cationic polymer linear polyethylenimine (593215, Polysciences). Vero and vT2 cells at approximately 40–60% confluence were transfected with mixtures of plasmid DNA and jetPRIME transfection reagent (101000046, Polyplus). The cell lines used in this study tested negative for mycoplasma contamination using the Mycolor One-step Mycoplasma Detector Kit (Vazyme) and were authenticated by short tandem repeat analysis.

## Immuno- and affinity-precipitations

For the antibody-based immunoprecipitation assay, cells were harvested and lysed in lysis buffer (50 mM Tris-HCl [pH 7.5], 150 mM NaCl, 5 mM EDTA, 0.2% NP40) containing protease inhibitor cocktail (Sigma, P8340) at 4°C for 1 hr. After a few seconds of sonication using a micro-tip and centrifugation at 12,000 × $g$ for 10 min at 4°C, supernatants were incubated with primary antibodies and Protein A/G magnetic beads (Thermo Scientific, 88803) overnight at 4°C. The samples were then washed five times with wash buffer (50 mM Tris-HCl [pH 7.5], 150 mM NaCl, and 0.2% NP40). Precipitates were eluted using IgG elution buffer (Thermo Scientific, 21028).

For Flag-based immunoprecipitation and CBD-, Strep- and s-tag-based affinity precipitations, cells in 10 cm dishes were transfected with appropriate expression vectors. 48 hr later, cells were harvested and lysed at 4°C in lysis buffer (150 mM NaCl, 5 mM EDTA, 50 mM Tris-HCl [pH 7.5], and 0.2% NP40) containing protease inhibitor cocktail (Sigma, P8340) for 1 hr. After incubation, the samples were sonicated using a micro-tip for several seconds. Lysates were then centrifuged at 12,000 × $g$ for 10 min at 4°C. The supernatants were harvested and mixed with Flag-antibody beads (Sigma, catalog number M8823), chitin resin (New England Biolabs, S6651), Strep-Tactin resin (Zymo Research, catalog number P2004-1-5), or S-protein agarose (Novagen, 69704). Subsequently, the mixtures were incubated at 4°C overnight with gentle shaking. Precipitates were then washed three to five times with wash buffer (50 mM Tris-HCl [pH 7.5], 150 mM NaCl, and 0.2% NP40). Coprecipitated proteins were released from the beads by heating/denaturation (CBD-, s-AP) in 1×loading buffer or eluting with Strep-Elution buffer (Strep-AP, Zymo Research, P2004-3-30) or lysis buffer containing 3xFlag peptide (Flag IP, Sigma, catalog number F4799).

For denatured precipitation-immunoblot analysis of ubiquitination, cells harvested from the dishes were treated with 300 µl of PBS containing 1% SDS and boiled for 10 min. Subsequently, lysates were diluted 1:3 with PBS and sonicated using a micro-tip sonicator for 1 min. After centrifugation at 12,000 × $g$ for 10 min, the supernatants were harvested and diluted five times in PBS. Supernatants were incubated with Flag-antibody beads, Strep-Tactin resin, or S-protein agarose at 4°C overnight. Finally, the beads and associated proteins were washed in lysis buffer for five times and proteins were released by heating/denaturation (CBD-, s-AP) in 1× loading buffer or eluted with Strep-Elution buffer (Strep-AP) or lysis buffer containing 3×Flag peptide (Flag IP).

For urea-washed precipitation–immunoblot analysis of extracellular proteins, culture medium from cells in 10 cm dishes transfected with appropriate expression vectors was harvested and incubated

with 0.2% NP40, protease inhibitor cocktail and respective Flag-antibody beads, namely chitin resin, Strep-Tactin resin, or S-protein agarose. Then, the mixtures were incubated at 4°C overnight with gentle shaking. Precipitates were then washed three to five times with wash buffer (PBS containing 0.2% NP40) and were released from beads by heating/denaturation (CBD-, s-AP) in 1× loading buffer or eluting with Strep-Elution buffer (Strep-AP) or lysis buffer containing 3xFlag peptide (Flag IP). Finally, the samples were subjected to immunoblot or dot blot analysis.

## Trichloroacetic acid/acetone precipitation, proteolysis, and mass spectrometry analysis

For the structural protein interactome identification (see *Figure 1—figure supplement 1A-C*), cells from dishes were transfected with Flag-2Strep tagged expression vectors. 48 hr later, cells were harvested and lysed. Double precipitations (Strep AP and Flag IP) were carried out as described above. The precipitated samples in elution buffer (PBS containing 3×Flag peptide) were then subjected to reduction, alkylation, and trypsin digestion. Subsequently, the peptides were analyzed by LC–MS on an Orbitrap Fusion Lumos mass spectrometer (Thermo Fisher) and the data were The precipitated samples in elution buffer (PBS containing 3×Flag peptide) were reduced with 50 mM Dithiothreitol in 10 mM ammonium bicarbonate (ABC) at 60°C for 45 min followed by alkylating with 100 mM chloroacetamide in 10 mM ABC at room temperature in the dark for 15 min. MS interfering reagents were removed by precipitating proteins by adding 8 volumes of 10% trichloroacetic acid (TCA) in cold acetone at −20°C for 2 hr. The pellet was centrifuged at 16,000 × $g$ for 10 min at 4°C. The TCA/acetone supernatant was removed, and the protein pellet was washed with an equivalent 8 volumes acetone at −20°C for 10 min prior to centrifuging at 16,000 × $g$ for 10 min at 4°C. The acetone supernatant was removed from the protein pellet.

The TCA/acetone protein pellets were resuspended and digested overnight at 37°C in 100 µl 100 mM TEAB with 0.3 µg Trypsin/Lys-C (Pierce, A41007) per sample. Each sample was acidified by adding 25 µl of 5%TFA, desalted by solid phase extraction using Oasis HLB Sorbent (Waters, 186001828BA) and dried by vacuum centrifugation.

Peptides were resuspended in 20 µl 0.1% formic acid and analyzed on an Orbitrap-Fusion Lumos (Thermo Fisher Scientific) interfaced with an Easy-nLC1200 UPLC by reversed-phase chromatography using a 2–90% acetonitrile in 0.1% formic acid gradient over 90 min at 300 nl/min on a 75 µm × 150 mm ReproSIL-Pur-120-C18-AQ column 3 µm, 120 Å. Eluting peptides were sprayed into the mass spectrometer through a 1-µm emitter tip (New Objective) at 2.4 kV. Survey scans (MS) of precursor ions were acquired from 370 to 1700 *m/z* at 120,000 resolution at 200 *m/z*, with a 4e5 automatic gain control (AGC) target and maximum injection time (IT) set to auto. Precursor ions with charge states 2–7 were individually isolated within 0.7 *m/z* by data dependent monitoring and 10 s dynamic exclusion, and fragmented using an HCD activation collision energy 31. Fragmentation spectra (MS/MS) were acquired using a 5e5 AGC, auto maximum IT at 60,000 resolution.

## Data analysis

Fragmentation spectra were processed by Proteome Discoverer v2.5 (PD2.5, Thermo Fisher Scientific) and searched with Mascot v.2.8.0 (Matrix Science, London, UK) against RefSeq2021_204_H_sapiens_220315.fasta; SARS_CoV_2_20220329.fasta; SARS-CoV-2_20210204 database with 116177 entries. Search criteria included trypsin enzyme, one missed cleavage, 3 ppm precursor mass tolerance, 0.015 Da fragment mass tolerance, with carbamidomethyl of cysteine was specified in Mascot as a fixed modification. Deamidated of asparagine and glutamine, oxidation of methionine, phospho of serine and threonine and GG of lysine were specified in Mascot as variable modifications. Peptide identifications from the Mascot searches were validated with Scaffold (version 5.1.2, Proteome Software Inc, Portland, OR). Peptide identifications were accepted if they could be established at greater than 95.0% probability by the Peptide Prophet algorithm with Scaffold delta-mass correction. Protein identifications were accepted if they could be established at greater than 95.0% probability and contained at least 1 identified peptide. Protein probabilities were assigned by the Protein Prophet algorithm. Proteins that contained similar peptides and could not be differentiated based on MS/MS analysis alone were grouped to satisfy the principles of parsimony.

## ProteomeXchange accession code

The mass spectrometry proteomics data have been deposited to the ProteomeXchange Consortium via the PRIDE (*Perez-Riverol et al., 2022*) partner repository with the dataset identifier **PXD049221**.

## Immunofluorescence assay

For immunofluorescence analysis, cells were transfected with appropriate expression vectors. 24 hr later, cells were washed twice with PBS and fixed with 4% paraformaldehyde (PFA) in PBS for 30 min at room temperature. Fixed cells were then washed three times with PBS, permeabilized and blocked with 0.3% Triton X-100 in blocking buffer (1× PBS, 5% BSA) for 1 hr at room temperature. Cells on the slides were then incubated with primary antibody overnight at 4°C. After three times washing, the cells were incubated with appropriate fluorescently conjugated secondary antibodies for 1 hr at room temperature. Coverslips were mounted with ProLong Diamond Antifade Mountant with DAPI (Thermo Fisher, P36971).

## Proximity ligation assays

PLAs were carried out using the mouse/rabbit Duo-link In Situ Red Kit from Millipore-Sigma-Aldrich (catalog numbers DUO92002, DUO92004, and DUO92008), according to the manufacturer's instructions. In brief, HEK293 or vT2 cells were seeded on 8-well glass chamber slides and fixed in PBS containing 4% PFA for 15 min at room temperature. Cells were then blocked in Blocking Solution (37°C for 1 hr) prior to incubation with primary antibodies in antibody dilution buffer (4°C overnight). After incubation with PLA probe solution (37°C, 1 hr), ligation solution (37°C, 30 min), and amplification solution (37°C, 100 min.), and rinsing with 1× and then 0.01× Wash Buffer B, the slides were mounted with a coverslip using ProLong Diamond Antifade Mountant with DAPI. The red dot signals were observed and photographed using a Leica SP8 confocal microscope.

## Lentivirus production

HEK293 cells in 15 cm dishes at ~80% confluency were co-transfected with 12 µg (each) lentiviral vector together with 9 µg psPAX2 and 3 µg pMD2.G (Addgene, 12260 and 12259, respectively). 6 hr later, cultural media were replaced by fresh media. 48 hr later, the media were harvested and centrifuged at 25,000 rpm for 2 hr at 4°C. The supernatant was discarded and 2 ml of DMEM medium were added into the viral pellets and kept at 4°C overnight. Virus pellets were resuspended and aliquots were stored at –80°C.

## Cell line construction

The vector lentiCRISPRv2-Blast (blasticidin-selectable) was obtained from Addgene (catalog number 98293) for the construction of ITCH knockout cell lines. The CRISPR-Blast vector was used for double-stranded (ds) oligonucleotides cloning to generate ITCH gene-targeting gRNA; ITCH gRNA sequences were cloned into the BsmBI site of lentiCRISPRv2-Blast. The sequences, including lentiviral vector cloning-compatible flanking nucleotides, were as follows: F: <u>CACCG</u>GAACGGCGGGTTGACAACAT , R: <u>AAAC</u>ATGTTGTCAACCCGCCGTT<u>CC</u>. LentiCRISPRv2-Blast and LentiCRISPRv2-Blast-ITCH-based lentiviruses were, respectively, incubated with HEK293 and vT2 cells in the presence of polybrene (5 µg/ml). After 48 hr of incubation, the infected cells were treated with blasticidin (10 µg/ml) to select lentivirus-positive cells. Then, clonal cell lines were derived by limiting dilution and culturing in 96-well plates. Finally, the clonal cell line candidates were verified by immunoblot blotting and sequencing. For the construction of the p62 knockdown cell line, predesigned shRNA clones TRCN0000007237 (Target Sequence: CCTCTGGGCATTGAAGTTGAT) and TRCN0000431509 (Target Sequence: GAGG ATCCGAGTGTGAATTTC) were ordered from Millipore Sigma. Lentiviruses were produced and then incubated with HEK293 cells in the presence of polybrene (5 µg/ml). After 48 hr incubation, the infected cells were treated with puromycin (3 µg/ml) for 1 week to select lentivirus-positive cells. Finally, the cell lines were tested by immunoblotting.

## SARS-CoV-2 replication assay

The vT2 cells were incubated with SARS-CoV-2 at various MOI. After 1 hr incubation, the virus culture medium was replaced by a fresh medium. After 48 hpi, the cells were imaged and then lysed with a loading buffer for further immunoblotting assays. Meanwhile, the culture medium was harvested for

virus titer and copy number assay. For the TCID50 assay, Vero E6 cells were seeded in 96-well plates. 18 hr later, a 10-fold serial dilution of the culture medium containing the SARS-CoV-2 virus was incubated with Vero E6 cells. Then, the cells were visualized at 48 hpi. SARS-CoV-2 virus copy numbers were determined by N-directed qRT-PCR (N1 Forward Primer: GACCCCAAAATCAGCGAAAT, N1 Reverse Primer: TCTGGTTACTGCCAGTTGAATCTG). In brief, viral RNA was extracted from a culture medium using Quick-RNA Viral Kit (ZYMO research, R1035). The viral cDNA was generated using the Verso cDNA Synthesis Kit (Thermo Fisher, AB-1453/B). For quantitative PCR, reactions were performed in a 96-well microplate with SYBR green 2x master mix (Thermo Fisher, A25742). FP-SARS-N (Addgene, #153201) was used as the positive control for copy number calculation.

## Antibodies

The following primary antibodies were used: Flag (Sigma, F1804); Flag (Sigma, F3165); Flag (Cell Signaling Technology, 14793S); GAPDH (Invitrogen, MA5-27912); β-tubulin (Cell Signaling Technology, 2128S); s (BioLegend, 688102); Strep (Invitrogen, MA5-17283); CBD (New England BioLabs, E8034S); ubiquitin (Cell Signaling Technology, 58395S); K63-linkage-specific antibody (Enzo Life Sciences, BML-PW0600-0100); K48-linkage-specific antibody (Cell Signaling Technology, 8081S); Spike (Proteintech, 28867-1-AP); M (Proteintech, 28882-1-AP); E (Proteintech, 28904-1-AP); ITCH (Santa Cruz, sc-28367); ITCH (Novus Biologicals, NB100-68142); p62 (Cell Signaling Technology, 88588S and 7695S); GM130 (Proteintech, 11308-1-AP); LAMP1 (Cell Signaling Technology, 9091S); OPTN (Cayman Chemical, 100002); LC3 (Proteintech, 14600-1-AP); LC3B (Cell Signaling Technology, 3868S); SARS-CoV-2 Membrane protein (Cell Signaling Technology, 15333S); SARS-CoV-2 Envelope protein (Cell Signaling Technology, 74698S); furin (Proteintech, 18413-1-AP); Cathepsin L (Proteintech, 10938-1-AP); NDP52 (Proteintech, 12229-1-AP); NBR1 (Proteintech, 16004-1-AP); FAM134B (Proteintech, 21537-1-AP); RTN3 (Proteintech, 12055-2-AP), and NIX (Proteintech, 12986-1-AP). Horseradish peroxidase-conjugated secondary antibodies were as follows: ECL anti-rabbit IgG, HRP-linked antibody (Cell Signaling Technology, 7074S), anti-mouse IgG, HRP-linked antibody (Cell Signaling Technology, 7076S), anti-Rabbit IgG HRP (Rockland, 18-8816-31); anti-Mouse Ig HRP (Rockland, 18-8817-30); and anti-Rabbit IgG (Vazyme, RA1008-01). Secondary antibodies used in immunofluorescence analysis: Donkey anti-Mouse IgG (H+L) Highly Cross-Adsorbed Secondary Antibody, Alexa Fluor 488 (Invitrogen, A-21202); Donkey anti-Rabbit IgG (H+L) Highly Cross-Adsorbed Secondary Antibody, Alexa Fluor 555 (Invitrogen, A-31572); Donkey anti-Goat IgG (H+L) Cross-Adsorbed Secondary Antibody, and Alexa Fluor 647 (Invitrogen, A-21447).

## Statistical analysis

Statistical analyses were conducted using unpaired two-sided Student's $t$-tests for two-group comparisons and one-way ANOVA with the Tukey post hoc test for multiple group comparisons, utilizing GraphPad Prism software. The sample size '$n$' denotes the number of biological replicates unless otherwise indicated. Data are represented as mean ± s.d. p values less than 0.05 were considered statistically significant.

## Acknowledgements

This work was supported by National Institutes of Health (NS074324), the Walder Foundation, and Johns Hopkins University. We would like to thank members of Wang laboratory for helpful discussion and thank Dr. John Nicholas for providing vectors and revising the manuscript.

## Additional information

### Funding

| Funder | Grant reference number | Author |
|---|---|---|
| National Institutes of Health | NS074324 | Jiou Wang |
| Walder Foundation | | Jiou Wang |

| Funder | Grant reference number | Author |
| --- | --- | --- |
| Johns Hopkins University | | Jiou Wang |

The funders had no role in study design, data collection, and interpretation, or the decision to submit the work for publication.

## Author contributions

Qiwang Xiang, Conceptualization, Data curation, Formal analysis, Validation, Investigation, Visualization, Methodology, Writing – original draft, Writing – review and editing; Camille Wouters, Peixi Chang, Haley Heine, Data curation, Investigation; Yu-Ning Lu, Validation, Investigation; Mingming Liu, Data curation; Haocheng Wang, Sunning Qian, Junqin Yang, Investigation; Andrew Pekosz, Yanjin Zhang, Validation, Writing – review and editing; Jiou Wang, Conceptualization, Resources, Supervision, Funding acquisition, Validation, Visualization, Methodology, Writing – original draft, Project administration, Writing – review and editing

## Author ORCIDs

Qiwang Xiang ![ORCID] https://orcid.org/0009-0001-4710-7572
Andrew Pekosz ![ORCID] https://orcid.org/0000-0003-3248-1761
Yanjin Zhang ![ORCID] https://orcid.org/0000-0002-5847-3260
Jiou Wang ![ORCID] https://orcid.org/0000-0001-9115-8708

Reviewer #1 (Public review): https://doi.org/10.7554/eLife.105105.3.sa1
Author response https://doi.org/10.7554/eLife.105105.3.sa2

## Additional files

### Supplementary files

MDAR checklist

### Data availability

All data needed to evaluate the conclusions in the paper are present in the paper and/or the Supplementary Materials.

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
