## [Editor Report · eLife Assessment]

This **valuable** study demonstrates that the E3 ligase ITCH regulates several steps of the SARS-CoV-2 replication cycle by enhancing ubiquitination of viral envelope and membrane proteins. The phenotypic data are based on **solid** evidence showing a role for ITCH in distinct phases of viral replication and host processes. The findings lay the ground work for future studies to decipher detailed molecular mechanisms that explain how ITCH regulates SARS-CoV-2.

---

## [Referee Report · Reviewer #1 (Public review)]

Summary:

The authors investigated the role of an E3 ubiquitin ligase ITCH in regulating the viral life cycle of SARS-CoV-2. The authors showed that ITCH mediates ubiquitination of the membrane (M) and envelope (E) proteins of SARS-CoV-2. Ubiquitination of E and M result in enhanced interactions between the structural proteins and redistribution of the structural proteins into autophagosomes. The authors claim that the enhanced interactions between structural proteins and trafficking of the structural proteins into autophagosomes contribute to SARS-CoV-2 replication and egress, prompting ITCH as a potential antiviral target. ITCH also alters the cellular distribution of host proteases important for spike cleavage which protect and stabilize spike with cleavage. The authors also demonstrated that SARS-CoV-2 replication is augmented by ITCH in which virus replication is significantly impaired in cells lacking ITCH expression.

Strengths:

The authors provided high quality data with appropriate experimental controls to justify their claims and conclusions. The mechanistic analyses are excellent and presented in a logical manner. The investigation of the role of ubiquitination in coronavirus assembly and egress is novel as most previous studies focused on its role in mediating innate immune responses.

Comments on revisions:

The authors have addressed my previous concerns.

---

## [Author Response]

The following is the authors’ response to the original reviews.

**Public Reviews:**

**Reviewer #1 (Public review):**
Summary:The authors investigated the role of an E3 ubiquitin ligase ITCH in regulating the viral life cycle of SARS-CoV-2. The authors showed that ITCH mediates ubiquitination of the membrane (M) and envelope (E) proteins of SARS-CoV-2. Ubiquitination of E and M results in enhanced interactions between the structural proteins and redistribution of the structural proteins into autophagosomes. The authors claim that the enhanced interactions between structural proteins and trafficking of the structural proteins into autophagosomes contribute to SARS-CoV-2 replication and egress, prompting ITCH as a potential antiviral target. ITCH also alters the cellular distribution of host proteases important for spike cleavage which protect and stabilize spike with cleavage. The authors also demonstrated that SARS-CoV-2 replication is augmented by ITCH in which virus replication is significantly impaired in cells lacking ITCH expression.Strengths:The authors provided high-quality data with appropriate experimental controls to justify their claims and conclusions. The mechanistic analyses are excellent and presented in a logical manner. The investigation of the role of ubiquitination in coronavirus assembly and egress is novel as most previous studies focused on its role in mediating innate immune responses.Weaknesses:Although the authors showed that ITCH ubiquitinates E and M proteins, the claim that such ubiquitination promotes virion assembly and egress is circumstantial. The enhanced interaction between the structural proteins and targeting of ubiquitinated structural proteins into autophagosomes does not necessarily result in increased virion production and release as suggested by the authors. There is a disconnect between the ubiquitination of structural proteins and the role of ITCH in augmenting virus replication as shown in Fig. 6A and B. In addition, the authors showed that the catalytic activity of ITCH is important for the localization and maturation of host proteases. However, the mechanism behind is unknown. Also, it is unclear how protection of spike from cleavage conferred by ITCH explains its role in promoting replication as a lack of spike cleavage would inevitably compromise entry. The major weakness of the manuscript is the lack of experimental data that explains the molecular role of ITCH in relation to its phenotype observed during SARS-CoV-2 infection.

We sincerely thank the reviewer for the positive evaluation of the quality, rigor, and novelty of our study. We particularly appreciate the thoughtful comments regarding the mechanistic link between ITCH-mediated ubiquitination and viral assembly/egress, as well as the broader implications for SARS-CoV-2 replication.

Our data support a model in which ITCH-mediated ubiquitination of the structural proteins M and E enhances their interactions and promotes their trafficking into autophagosomal compartments, ultimately contributing to increased virion production and release. The phenotypic outcomes observed in Fig. 6A-B (replaced by re-measured viral infectious titer and genomic copy number in the culture medium of vT2-WT and vT2-KO cells) are consistent with our earlier findings in Figs. 1-5, which demonstrate that ITCH promotes SARS-CoV-2 replication. Thus, the replication defect observed in ITCH-deficient cells aligns with the mechanistic effects of ITCH on structural protein ubiquitination and trafficking.

We agree with the reviewer that directly linking ubiquitination of structural proteins to virion production would further strengthen the mechanistic connection. However, direct detection of ubiquitinated virions in vitro, particularly by electron microscopy (EM), remains technically challenging. Our laboratory has not yet established an EM-based platform optimized for high-resolution SARS-CoV-2 virion analysis. Furthermore, it is possible that ubiquitin chains conjugated to structural proteins are cleaved during or after virion egress, which would complicate their detection in released particles. These technical and biological considerations currently limit direct visualization of ubiquitinated virions.

Regarding the role of ITCH in regulating the localization and maturation of host proteases, our recent studies [1, 2] have demonstrated that ITCH is involved in Golgi fragmentation, leading to altered furin distribution and impaired cathepsin L maturation. These findings provide mechanistic insight into how ITCH catalytic activity may influence host protease processing. We have incorporated this discussion into the revised manuscript (last paragraph of the Discussion section) to better contextualize our observations.

With respect to spike cleavage, although S1/S2 processing is required for SARS-CoV-2 entry, accumulating evidence suggests that excessive intracellular cleavage may be detrimental to virion stability. For example, in Vero cells lacking TMPRSS2, virions containing cleaved S1 and S2 are less stable [3]. Additionally, the D614G substitution renders the spike protein more resistant to cleavage, reduces S1 shedding, and enhances incorporation of intact spike into virions, thereby increasing infectivity and stability [4-6]. These findings suggest that maintaining intact spike during intracellular assembly may be advantageous for the viral life cycle. In this context, ITCH-mediated modulation of host protease distribution and spike processing may help preserve spike integrity within assembling virions.

Taken together, the ability of ITCH to (i) enhance structural protein interactions, (ii) facilitate trafficking through autophagosomal pathways, and (iii) promote incorporation of intact spike into virions provides a coherent mechanistic framework explaining how ITCH enhances virion production and release. While additional studies will be required to further dissect the precise molecular details, our data collectively support a functional link between ITCH ubiquitin ligase activity and SARS-CoV-2 assembly and egress.

**Reviewer #2 (Public review):**
Summary:In this manuscript Qiwang Xiang et al. investigated the role of the E3 ubiquitin ligase ITCH in the life cycle of SARS-CoV-2. They claim the following:(i) ITCH promotes virion assembly by interacting with E and M proteins and enhancing their K63-linked ubiquitination(ii) ITCH-mediated ubiquitination promotes autophagosome-dependent secretion of viral particles.(iii) ITCH stabilizes the viral spike protein by impairing its processing by furin and catepsin L proteases.The manuscript provides an interesting exploration of ITCH's role in the SARS-CoV-2 life cycle but requires additional work to strengthen key claims and address potential confounding factors.Strengths:The experiments are sufficiently clear in documenting that ITCH activity is critical for efficient SARS-CoV-2 replication and for M and E proteins K63-linked ubiquitinationWeaknesses:The manuscript does not convincingly demonstrate how ITCH-mediated ubiquitination of E and M impacts virus assembly and release. Identifying the specific lysine residues in M and E targeted by ITCH, and generating mutant VLPs or recombinant viruses, would strengthen the conclusions.Most of the conclusions rely on ITCH overexpression data, which may have off-target effects on Golgi integrity and vesicular trafficking. For instance, figure 4F provides evidence of altered Golgi morphology and TGN46 fragmentation raising concerns that ITCH overexpression could indirectly mislocalize furin, affecting S1/S2 cleavage of the spike protein. In addition, inhibition of furin activity may also lead to off-target effects, given its role in processing numerous host proteins.Similarly, ITCH overexpression is likely to indirectly affect cathepsin-L maturation. In addition, the manuscript does not clarify how impaired cathepsin L activity would influence virus assembly or release.A major concern is also the lack of quantification and statistical analysis of immunofluorescence images throughout the manuscript, which undermines the reliability of these observations.

We sincerely thank the reviewer for recognizing the importance of ITCH in SARS-CoV-2 replication and for the constructive and insightful suggestions to further strengthen the manuscript.

Regarding the impact of ITCH-mediated ubiquitination of E and M on virus assembly and release, our data support a model in which ITCH promotes K63-linked ubiquitination of the E and M proteins, facilitating their recruitment to p62-positive autophagosomal compartments. This recruitment likely enhances the spatial proximity and interaction frequency of structural proteins within assembly sites, thereby promoting efficient virion assembly and subsequent release via autophagosome-dependent secretory pathways.

We agree that identifying the specific lysine residues in M and E targeted by ITCH and generating mutant VLPs or recombinant viruses would provide a more direct mechanistic link. These are important and technically demanding experiments that require extensive mutagenesis and reverse genetics approaches. While beyond the scope of the current study, we fully acknowledge their value and plan to pursue these directions in future work to further refine the mechanistic understanding of ITCH-dependent ubiquitination during coronavirus assembly.

Regarding the reliance on ITCH overexpression systems, we acknowledge the reviewer’s concern that ectopic ITCH expression may affect Golgi integrity and vesicular trafficking. Indeed, our recent studies [1, 2] demonstrate that ITCH catalytic activity disrupts Golgi structure, resulting in altered furin distribution and impaired cathepsin L maturation. These findings provide mechanistic context for the phenotypes observed in the present study and suggest that ITCH regulates host protease localization through defined cellular pathways rather than nonspecific overexpression artifacts. We have now expanded the Discussion section (last paragraph) to clarify this mechanistic framework.

Importantly, SARS-CoV-2 infection itself significantly activates endogenous ITCH, and therefore our ectopic expression system likely mimics infection-induced ITCH activation rather than representing a purely artificial condition. In addition, key phenotypes, such as reduced viral replication and altered structural protein behavior, are consistently observed in ITCH-deficient cells, supporting the physiological relevance of ITCH activity in the viral life cycle.

Regarding cathepsin L (CTSL) maturation, we have expanded the Discussion to clarify how impaired CTSL activity may influence viral assembly and egress. ITCH inhibits CTSL maturation, thereby reducing excessive spike cleavage into smaller fragments. Although CTSL-mediated spike processing facilitates genome release following endocytosis [7, 8], CTSL is a lysosomal protease, and lysosomes are exploited by β-coronaviruses as egress organelles [9]. Excessive lysosomal proteolysis may therefore compromise virion integrity during egress. In this context, ITCH-mediated inhibition of CTSL maturation may preserve spike stability within assembling or trafficking virions, thereby promoting the production and release of infectious particles during the replication phase.

Regarding quantification and statistical analysis of immunofluorescence data, we appreciate this important point. In the revised manuscript, we have included expanded image panels with increased cell numbers, quantitative colocalization analyses to enhance the rigor of these observations.

**Reviewer #3 (Public review):**
Summary:Xiang et al. investigated the role of ubiquitin E3 ligase ITCH in SARS-CoV-2 replication. First, they described the role of ITCH on the structural proteins. Here, the ubiquitination of E and M (but not S) leads to an enhanced interaction and presumably virion assembly. In addition, E and M ubiquitination seems to be necessary for p62-guided sequestration into autophagosomes for secretion. Furthermore, ITCH regulates S proteolytic cleavage by changing furin localization and inhibiting CTSL protease maturation. In addition, SARS-CoV-2 infection upregulates ITCH phosphorylation, whereas knockout of ITCH reduces SARS-CoV-2 replication.Strengths:The proposed study is of interest to the virology community because it aims to elucidate the role of ubiquitination by ITCH in SARS-CoV-2 proteins. Understanding these mechanisms will address broadly applicable questions about coronavirus biology and enhance our knowledge of ubiquitination's diverse functions in cell biology.Weakness:The involvement of ubiquitin ligases in SARS-CoV-2 replication is not entirely new (see E3 Ubiquitin Ligase RNF5; Yuan et al., 2022; Li et al., 2023). While the data generally support the conclusions, additional work is needed to confirm the role of ITCH in SARS-CoV-2 replication in a biologically relevant context. The vast majority of data is based on transient overexpression experiments of ITCH, which ultimately leads to massive ubiquitination of several viral and host cell factors, including potentially low-affinity substrates not typically recognized under physiological conditions. In addition to that, nearly all experiments were done in cells co-overexpressing ITCH and the viral structural proteins (or cellular proteases) in HEK293T cells. Therefore, a proteomic analysis of protein ubiquitination in (a) SARS-CoV-2-infected cells (ideally several cell types) and (b) SARS-CoV-2-infected v2T-ITCH-KO cells would verify the ITCH-related ubiquitination of e.g., E and M and would strengthen the whole manuscript. In addition, the few key experiments using SARS-CoV-2 infected cells were performed in VeroE6 cells, which are neither human nor lung-derived. Only in one experiment were lung-derived Calu3 cells included.Moreover, the manuscript names ITCH as a central regulator of SARS-CoV-2 replication. If ITCH is beneficial for E and M interaction and thereby aids virion assembly, showing its effect on VLP production would be desirable. Clarifications regarding data acquisition and data analysis could strengthen the manuscript and its conclusions.

We sincerely thank the reviewer for the thoughtful evaluation and for highlighting the importance of demonstrating physiological relevance.

We agree that the involvement of E3 ubiquitin ligases in SARS-CoV-2 replication is not entirely unprecedented. Accordingly, we have expanded the Introduction to discuss RNF5 and other E3 ligases previously implicated in SARS-CoV-2 biology (e.g., Yuan et al., 2022; Li et al., 2023), thereby clarifying how ITCH differs mechanistically.

Regarding the reliance on transient overexpression systems, we acknowledge the reviewer’s concern. Importantly, SARS-CoV-2 infection itself significantly induces ITCH phosphorylation and activation. Therefore, our ectopic expression system likely mimics infection-driven ITCH activation rather than representing a purely artificial condition. Moreover, key findings, including reduced viral replication and diminished E/M ubiquitination, were validated in ITCH knockout cells, supporting the physiological relevance of ITCH-dependent structural protein ubiquitination under endogenous conditions.

We appreciate the suggestion to perform a global proteomic analysis of ubiquitinated proteins in (i) SARS-CoV-2-infected cells and (ii) SARS-CoV-2-infected ITCH-KO cells. Such analyses would indeed provide a comprehensive and unbiased assessment of ITCH-dependent ubiquitination events. While this approach is beyond the scope of the current study, we fully recognize its value and plan to pursue it in future investigations to further refine the mechanistic understanding of ITCH-mediated ubiquitination during coronavirus assembly.

With respect to the cellular models used, Vero E6/TMPRSS2 cells are widely established for SARS-CoV-2 propagation due to their robust viral replication, rapid growth, and reduced culture-adapted mutations. Compared with Calu-3 cells, which grow more slowly and may acquire specific adaptations in certain viral genes during prolonged passage, Vero E6/TMPRSS2 cells maintain high viral stability and reproducibility, making them suitable for mechanistic studies. Nevertheless, we agree that human lung-derived systems are highly relevant, and we have included Calu-3 cell data where feasible to support translational relevance.

Regarding the role of ITCH in virion assembly, our data in Fig. 2 demonstrate that ITCH-mediated K63-linked ubiquitination enhances the interaction between E and M proteins, supporting a functional role in virus-like particle (VLP) formation. We agree that direct visualization and quantification of VLP production by EM would further strengthen this conclusion. Such experiments require additional optimization and will be pursued in future work to provide more direct structural evidence.

Finally, in response to the reviewer’s comments on data acquisition and analysis, we have expanded image panels, increased the number of quantified cells, and included quantitative colocalization analyses with appropriate statistical evaluation in the revised manuscript to enhance rigor and reproducibility.

**Recommendations for the authors:**

**Reviewer #1 (Recommendations for the authors):**
(1) The authors should compare the infectivity of SARS-CoV-2 generated in cell lines expressing or lacking ITCH to investigate the effects of ITCH on infectivity, possibly by measuring RNA to PFU ratio and determining the S cleavage pattern in purified virions.

We re-measured the viral infectious titer and genomic copy number in the culture medium of vT2-WT and vT2-KO cells infected at an MOI of 0.0001 for 24 h. ITCH ablation reduced the viral copy number by approximately 8-fold (Fig. 6B), while the infectious titer (TCID_₅₀_) decreased by at least 25-fold (Fig. 6A), indicating that loss of ITCH markedly impairs the formation of infectious viral particles. This finding is consistent with the role of ITCH in promoting Spike (S) protein cleavage.

As suggested, to assess the S cleavage pattern in secreted virions, we precipitated proteins from the culture medium of SARS-CoV-2–infected cells with or without ITCH expression. Analysis of the precipitated S proteins revealed that the loss of ITCH markedly altered the integrity of full-length S in SARS-CoV-2 virions (Fig. S7A).

(2) The authors should strengthen the connection between ubiquitination of structural proteins and viral egress by measuring infectious virus particles in the supernatants from cells with or without ITCH expression by plaque assay. However, this cannot be accurately achieved without performing the experiment described in point 1 as cleavage of spike and infectivity would affect the results.

While a plaque assay was not performed, we quantified infectious viral particles in the supernatants using the TCID_₅₀_ assay. These analyses showed that loss of ITCH resulted in a marked reduction in infectious virion production (>25-fold; Fig. 6A). In contrast, viral genomic copy numbers, which reflect both infectious and non-infectious particles, were reduced by approximately eightfold (Fig. 6B). The disproportionate reduction in infectious titer relative to viral copy number (approximately threefold difference) is consistent with a defect in virion infectivity, most likely due to impaired S cleavage in the absence of ITCH (Fig. S7A). The reduction in viral copy numbers suggests that ITCH-dependent ubiquitination of viral structural proteins contributes to efficient viral assembly and egress.

(3) The authors should strengthen the connection between ubiquitination of structural proteins and virion assembly by EM.

We appreciate the reviewer’s insightful comment. However, detecting ubiquitinated virions in vitro via electron microscopy (EM) remains technically challenging. At present, our laboratory has not yet established an EM-based system optimized for SARS-CoV-2 virion analysis. Moreover, it is also possible that ubiquitin chains present on virions may be cleaved during or after the viral egress process, further complicating their detection.

**Reviewer #2 (Recommendations for the authors):**
Supp. Figure 2: the authors should provide sequencing data for both ITCH-KO clones for consistency.

The sequence for both ITCH-KO clones have been included now (Fig. S2C).

Figure 2: All interaction data between structural proteins and p62 rely on ITCH overexpression. It would be helpful to include data in ITCH-KO cells as controls to validate these findings.

As suggested, we performed E-based immunoprecipitation in wild-type (WT) and ITCH-knockout (KO) cells and found that E pulled down less p62 in the absence of ITCH, confirming that ITCH-mediated ubiquitination of E facilitates its interaction with p62 (Fig. 3C).

Figure 3H: Verify the middle LC3B panel, as it does not match the merge panel. Please, correct any discrepancies.

We thank the reviewer for pointing out this error. Fig. 3H (now Fig. 3J) has been corrected accordingly.

Figure 4F: the labeling of the different panels seems incorrect.

We have corrected the figure labeling.

The authors should perform cell viability assays in clomipramine-treated cells. In addition, the authors should clarify whether clomipramine's antiviral effects depend on ITCH expression, given the comparable virus copy numbers in treated WT (Fig. S7B) and ITCH-KO cells (Fig. S7C)

We thank the reviewer for this helpful comment. As shown Author response image 1., while clomipramine (Clom) treatment for 48 hours resulted in a modest reduction in cell number compared with the DMSO control, no apparent cell death was detected under these conditions.

**Author response image 1. sa2fig1:** Vero-TMPRSS2 (A) or Vero-ITCH-KO (B) cells were treated with DMSO or chloroquine (Clo) for 48 h, and cell viability was assessed by calcein AM staining (n = 3).

**Reviewer #3 (Recommendations for the authors):**
Results:Fig.2A and 2E display controversial results with different outcomes depending on the used bait. In my opinion, in both approaches, the overexpressed ITCH should be able to ubiquitinate M and E (since they are co-expressed). However, the interaction of E and M is not affected by the overexpression of ITCH or ITCH-CS when E is used as a bait (Fig.2A). In contrast, the interaction of E and M is enhanced in the presence of overexpressed ITCH (Fig.2E), when M is used as a bait.

We thank the reviewer for pointing this out. It should be noted that the blots display only the major (un-ubiquitinated) bands of E and M. When M was used as the bait, more E (main band, un-ubiquitinated form) was co-precipitated in the presence of ectopically expressed ITCH. In contrast, when E was used as the bait, comparable levels of M (main band, un-ubiquitinated form) were detected regardless of ITCH expression. These results suggest that ubiquitin-modified M can bind more E, whereas ubiquitin-modified E does not significantly affect its interaction with M. A more detailed explanation has been added to the revised text.

Fig.3A+3F: The authors claim a reduced E secretion when ITCH-KO cells or shRNA-treated p62 cells are used. I believe an input loading control of the supernatant displaying an equal amount of e.g. BSA is missing.

In response to the reviewer’s suggestion, we have now included Coomassie Brilliant Blue (CBB) staining of the culture medium (now shown in Fig. 3A and Fig. 3F).

Fig.3B: ITCH does not interact with E (or M) alone in the displayed data. The data is comparable with data observed for the interaction with S (Supp.4A). However, the author claims that ITCH interacts with M and E but not S (page 11).

We would like to clarify that in ECL-based Western blotting, strong signals can mask weaker ones due to contrast limitations. In this experiment, ectopic expression of ITCH produced a strong signal that obscured the endogenous ITCH band. Upon longer exposure, the endogenous ITCH signal becomes visible. Additionally, our data presented in Fig. 1 and the new data in Fig. 3C demonstrate the interaction between the relevant proteins.

Fig 3F: A scrambled control is missing. Moreover, it would be desirable to see if overexpression of p62 would enhance E release to verify that ITCH ubiquitination and p62-positive autophagosomes are necessary for E release.

We appreciate the reviewer’s comment. Proteins in the culture medium were precipitated using TCA, and Coomassie Brilliant Blue (CBB) staining has been included (now shown in Fig. 3F). Additionally, E release was examined in the presence of overexpressed p62, and the results showed that p62 overexpression increased the level of E detected in the medium (now shown in Fig. 3G).

Fig.3: Overall, an experiment using, e.g. cycloheximide (protein synthesis inhibitor) and MG132 (proteasome inhibitor) would strengthen the hypothesis that E and M are not degraded in a lysosome after ITCH overexpression. In my opinion, a colocalization experiment with LAMP1 is unsuitable to draw this conclusion. Would the overexpression of a deubiquitinating enzyme diminish M, E and p62 interaction? Does ITCH/p62 only regulate the release of the overexpressed single E or M protein, or does it also affect VLP release? An experiment analyzing purified VLPs produced in ITCH- or ITCH-CS overexpressing cells would be desirable.

We thank the reviewer for these important questions. As suggested, we performed additional CHX and MG132 experiments. As shown in Fig. 3H and Fig. S3I, degradation of both E and M proteins was blocked by MG132 treatment, indicating that they are degraded via the proteasome pathway. Notably, MG132 treatment did not rescue the ITCH-mediated decrease of E/M levels, suggesting that the ITCH-dependent reduction of E and M is not mediated through the proteasome pathway. In addition, our recent back-to-back studies [1, 2] demonstrated that ITCH overexpression inhibits lysosomal function by impairing hydrolase maturation, suggesting that ITCH-mediated ubiquitination of E or M is unlikely to promote their degradation through the lysosomal pathway. Together, these data suggest that ITCH-mediated reduction of E and M is not due to enhanced degradation but is instead associated with their secretion.

Overexpression of deubiquitinating enzymes specifically targeting E or M (which remains to be identified) would likely reduce their interaction with p62.

Our data indicate that ITCH-mediated ubiquitination of E and M enhances their mutual interaction, supporting a role for this process in virus-like particle (VLP) formation. P62 would facilitate the release of VLPs by promoting the secretion of ubiquitinated E and M. In addition, the data presented in Fig. 2 indicate that ITCH enhances the mutual interaction of these structural proteins, thereby promoting virus-like particle (VLP) formation.

Fig.4A: PPC site mutation indicated in yellow. There is no yellow color.

We have revised the label to read “PPC site mutation indicated in red and green”.

Fig.4C: Why should the overexpression of ITCH or ITCH-CS affect the S protein cleavage when the cleavage site is anyhow mutated?

In this analysis, we aimed to verify that neither ITCH nor ITCH-CS affects the cleavage pattern of the mutated S protein. As these data are already presented in Fig. 4D (now Fig. 4C), the redundant result has been removed, and the corresponding description has been added to the revised text.

Fig.4C: Lysates from the single expression of S wt protein (-ITCH/ +ITCH-CS; as indicated in Fig.4B) is missing for comparison to S mut protein.

As these controls and related data are already presented in Fig. 4D (now Fig. 4C), the redundant result here has been removed.

Fig. 4D: Lane 5 and Lane 7 are labeled similarly. ITCH+ in Lane 5 needs to be removed.

We thank the reviewer for pointing out this error. The labeling (now Fig. 4C) has been corrected.

Fig 4G: A theoretical MOI of 1 does not lead to an infection of all cells. Therefore, including a third marker for infection control, e.g., N protein, would be helpful. This would clarify whether the changes in furin localization are due to infection.

We appreciate the reviewer for raising this point. Our goal was to examine whether SARS-CoV-2 infection affects the localization of furin (mouse antibody) relative to the Golgi marker (rabbit antibody). As suitable E, N, or M antibodies raised in goat or donkey were not available, we could not include those markers in this experiment. However, we did confirm M protein expression in parallel, and the infection efficiency was higher than 80% (Author response image 2.). To further validate that the observed changes in furin localization were due to viral infection, we have now included additional images showing a larger field of view containing more cells .

Fig.4: Generally, the colocalization of proteases with TGN46 should be analyzed quantitatively using, for example, Madner's overlap coefficient. This would be needed to draw the conclusion stated in the manuscript.

We appreciate the reviewer’s suggestion. We now have included the colocalization analysis in the Fig. 4E and F.

Fig.4/5: Overview IF pictures displaying additional cells would be desirable to clarify furin/cathepsin L localization in ITCH/ITCH-CS expressing cells. Otherwise, it looks (in my opinion) very subjective.

In response to the reviewer’s suggestion, we have included additional images with a larger field of view encompassing more cells for Fig. 4 and 5 (presented in Fig. S5B and S5H).

Fig.5D/G: MOI is missing in the figure legend.

As suggested, the MOI information has been added to the figure legend.

Fig.5D/G/6C/F: Infection control (e.g., N-protein) is missing in the Western Blots.

We have added the infection control M in the figures.

Fig.6: Why is the overall amount of ITCH reduced during the course of infection?

We appreciate the reviewer for raising this point. As shown in Fig. 6C and F, ITCH was significantly activated, as indicated by its phosphorylation at the T222 site during viral infection. This activation promotes ITCH self-ubiquitination.

Fig.6A: Would an overexpression of ITCH enhance viral replication?

Moderate upregulation of ITCH promotes viral replication, whereas excessive ITCH overexpression leads to cell death, which in turn partially reduces viral titers.

Discussion:Is there an explanation of how ITCH changes furin localization and CSTL maturation?

Our recent back-to-back studies[1, 2] demonstrated that ectopic ITCH expression disrupts Golgi integrity, resulting in altered furin distribution and impaired CSTL maturation. The relevant discussion has now been incorporated into the revised text (last paragraph of the Discussion section).

It would also be helpful to discuss the role of other known ubiquitin ligases like RNF5 in the replication of SARS-CoV-2 and other CoVs. Since the pandemic began, many interactome and host-factor studies in various cell types have been published. None of these studies identified ITCH so far. Could you comment on this?

As suggested, we have included additional known ubiquitin ligases involved in SARS-CoV-2 replication and in other viral systems (see the third paragraph of the Introduction).

Overall, in my opinion, the figure legends need to be improved. It is often not clear if ITCH is endogenously detected or overexpressed.

We thank the reviewer for the helpful suggestion. Additional details have been incorporated into the figure legends.

(1) Xiang Q, Lu Y, Wang H, Chen H, Chen P, Zhao X, et al. ITCH regulates Golgi integrity and proteotoxicity in neurodegeneration. Science Advances 2025; 11:eado4330.

(2) Xiang Q, Liu Y, Wang J. Golgi fragmentation driven by the USP11-ITCH axis triggers autolysosomal failure in neurodegeneration. Autophagy 2026.

(3) Peacock TP, Goldhill DH, Zhou J, Baillon L, Frise R, Swann OC, et al. The furin cleavage site in the SARS-CoV-2 spike protein is required for transmission in ferrets. Nature microbiology 2021; 6:899-909.

(4) Zhang L, Jackson CB, Mou H, Ojha A, Peng H, Quinlan BD, et al. SARS-CoV-2 spike-protein D614G mutation increases virion spike density and infectivity. Nature communications 2020; 11:1-9.

(5) Plante JA, Liu Y, Liu J, Xia H, Johnson BA, Lokugamage KG, et al. Spike mutation D614G alters SARS-CoV-2 fitness. Nature 2021; 592:116-21.

(6) Daniloski Z, Jordan TX, Ilmain JK, Guo X, Bhabha G, Sanjana NE. The Spike D614G mutation increases SARS-CoV-2 infection of multiple human cell types. Elife 2021; 10:e65365.

(7) Jaimes JA, Millet JK, Whittaker GR. Proteolytic cleavage of the SARS-CoV-2 spike protein and the role of the novel S1/S2 site. IScience 2020; 23:101212.

(8) Zhao M-M, Yang W-L, Yang F-Y, Zhang L, Huang W-J, Hou W, et al. Cathepsin L plays a key role in SARS-CoV-2 infection in humans and humanized mice and is a promising target for new drug development. Signal transduction and targeted therapy 2021; 6:1-12.

(9) Ghosh S, Dellibovi-Ragheb TA, Kerviel A, Pak E, Qiu Q, Fisher M, et al. β-Coronaviruses use lysosomes for egress instead of the biosynthetic secretory pathway. Cell 2020; 183:1520-35. e14.